# ROOT: Rethinking Offline Optimization as Distributional Translation via Probabilistic Bridge

**Manh Cuong Dao**[*]
National University of Singapore
cuongdao@nus.edu.sg

**The Hung Tran**[*]
Washington State University
hung.t.tran@wsu.edu

**Phi Le Nguyen**
Hanoi University of Science and Technology
lenp@soict.hust.edu.vn

**Thao Nguyen Truong**
National Institute of Advanced Industrial Science and Technology
nguyen.truong@aist.go.jp

**Trong Nghia Hoang**[†]
Washington State University
trongnghia.hoang@wsu.edu

## Abstract

This paper studies the black-box optimization task which aims to find the maxima of a black-box function using a static set of its observed input-output pairs. This is often achieved via learning and optimizing a surrogate function with that offline data. Alternatively, it can also be framed as an inverse modeling task that maps a desired performance to potential input candidates that achieve it. Both approaches are constrained by the limited amount of offline data. To mitigate this limitation, we introduce a new perspective that casts offline optimization as a distributional translation task. This is formulated as learning a probabilistic bridge transforming an implicit distribution of low-value inputs (i.e., offline data) into another distribution of high-value inputs (i.e., solution candidates). Such probabilistic bridge can be learned using low- and high-value inputs sampled from synthetic functions that resemble the target function. These synthetic functions are constructed as the mean posterior of multiple Gaussian processes fitted with different parameterizations on the offline data, alleviating the data bottleneck. The proposed approach is evaluated on an extensive benchmark comprising most recent methods, demonstrating significant improvement and establishing a new state-of-the-art performance. Our code is publicly available at https://github.com/cuong-dm/ROOT.

## 1 Introduction

Black-box optimization arises in scientific and engineering domains where evaluating each candidate solution is costly, often requiring extensive physical experiments or high-fidelity simulations [61]. For instance, designing energy-efficient hardware accelerators [5, 9, 37] involves numerous cycle-accurate

---

[*]These authors contributed equally.
[†]Corresponding authors: Manh Cuong Dao, The Hung Tran, Trong Nghia Hoang.

39th Conference on Neural Information Processing Systems (NeurIPS 2025).

simulations to assess configuration performance. In materials science, finding nanoporous structures with high adsorption capacity for carbon capture or hydrogen storage demands labor-intensive lab experiments [17, 20]. Similar challenges arise in protein design [21], molecular generation [16], and drug discovery [52], where evaluations are likewise expensive.

**Prior Literature.** Existing approaches to black-box optimization include both online and offline methods with the latter being an emerging alternative of the former. In particular, online methods such as Bayesian optimization [53, 34, 54, 73, 74, 62, 63, 64, 65, 33] have long been explored for black-box design tasks with provable performance guarantee in the asymptotic limit of data. However, their reliance on iterative experimentation makes them less practical in high-cost settings with limited to no experimentation budget. In contrast, black-box methods leverage past data to learn a surrogate model which can be used to find better designs without incurring new experimentation [36, 7, 18, 12, 27, 40, 59, 60, 14, 15]. In this paper, we focus on the offline setting, where the goal is to discover high-performing designs using only past experimentation data.

**Challenge.** Among offline methods, the main challenge is that surrogate models can become increasingly erroneous when the search moves away from the offline dataset, especially when offline data are biased or sparse, causing these models to overfit. To mitigate this issue, most existing approaches have focused on advancing techniques in (1) **forward modeling** that penalize high-value surrogate predictions at out-of-distribution (OOD) inputs [60, 13, 70], (2) **inverse modeling** that find most promising and reliable regions that contain high-performing inputs [40, 45, 39, 10] to sidestep the OOD issue of forward modeling, and (3) **search policies** that learn a direct plan to navigate from low-value inputs to high-value inputs [10, 38].

**Limitations.** Despite their promising results, forward and inverse approaches depend on learning a mapping (or inverse mapping) between input designs and their corresponding performance outputs using offline data. As a result, their effectiveness is inherently limited by the availability of data. Likewise, learning direct search policies also suffers from the same data bottleneck since these methods still need to sample heuristic trajectories from the offline dataset to use as learning feedback.

Furthermore, information regarding regions with high-performing inputs is often not observable from the offline data, especially in low-data scenarios, which might further restrict the effectiveness of the learned models/policies. To mitigate such data bottleneck, we propose to approach offline optimization from a *new perspective* of *distributional translation*, as highlighted below.

**Distributional Translation.** In essence, we view the offline data as an implicit distribution over low-value designs, and recast offline optimization as the task of learning a probabilistic transformation, or bridge, that transports this distribution toward a regime of higher-value inputs. By moving along the learned probabilistic bridge, we can reach regions of the input space associated with better designs. However, learning such a transformation is fundamentally limited by the scarcity of high-value examples. Our key insight to overcome this bottleneck is:

> **Although the feedback needed to learn such low-to-high probabilistic transformation is absent in the offline dataset, it can be derived from a distribution of synthetic functions that are similar to the (unknown) target function (up to a scale factor).**

These synthetic functions can be provably constructed across various output scales, alleviating the data bottleneck and broadening the solution scope of offline optimization.

**Technical Contributions.** Our solution perspective is substantiated via the following:

**1.** A general-purpose probabilistic bridge model that learns a direct mapping between two implicit data distributions. This perspective rethinks offline optimization through the lens of probabilistic transport. The resulting bridge can incorporate external guiding information from synthetic functions similar to the oracle to mitigate the data bottleneck. Once trained, it enables simulation of paths that move from low-value to high-value regimes (Section 3.1).

**2.** A pre-training and adaptation framework that (1) learns multiple Gaussian process priors [66] over synthetic functions resembling the target function, and (2) samples representative low- and high-value inputs from their corresponding closed-form mean functions. This generates high-quality training data that better delineates the low- and high-value regimes for learning the probabilistic bridge. The

intuition is that if the bridge can consistently map between these regimes across a wide range of functions similar to the oracle, it will be able to do the same for the oracle (Section 3.3).

**3.** An extensive empirical evaluation on a variety of benchmark datasets [59] and numerous existing baselines, establishing a new state-of-the-art performance, which significantly and consistently improves over previous work. Our empirical evaluation also features rich ablation studies examining in detail the practical impact of different components of our framework on its performance (Section 4).

## 2 Problem Definition and Preliminaries

This section provides a concise formulation of offline black-box optimization (Section 2.1) and important background on Gaussian processes (Section 2.2), which was used later for sampling additional data from synthetic functions similar to the oracle behind the offline data.

### 2.1 Offline Black-Box Optimization

Offline black-box optimization is formulated as the maximization of a black-box function $f(\boldsymbol{x})$ using only an offline dataset of observations $D_o = \{(\boldsymbol{x}_i, y_i)\}_{i=1}^n$ which $\boldsymbol{x}_i$ denote a past experiment design and $y_i = f(\boldsymbol{x}_i)$ is its corresponding evaluation. A direct approach to this problem is to learn a surrogate $g(\boldsymbol{x}; \boldsymbol{\omega}_*)$ of $f(\boldsymbol{x})$ via fitting its parameter $\boldsymbol{\omega}_*$ to the offline dataset,

$$\boldsymbol{\omega}_\star \quad \triangleq \quad \arg\min_{\boldsymbol{\omega}} L(\boldsymbol{\omega}) \quad \triangleq \quad \arg\min_{\boldsymbol{\omega}} \sum_{i=1}^n \ell\big(g(\boldsymbol{x}_i; \boldsymbol{\omega}), y_i\big) \ , \tag{1}$$

where $\boldsymbol{\omega}$ denotes a parameter candidate of the surrogate and $\ell(g(\boldsymbol{x}; \boldsymbol{\omega}), y)$ denotes the prediction loss of $g(.; \boldsymbol{\omega})$ on x if its oracle output is $y$. The (oracle) maxima of $f(\mathbf{x})$ is then approximated via,

$$\boldsymbol{x}_* \quad \triangleq \quad \arg\max_{\boldsymbol{x}} \ g(\boldsymbol{x}; \boldsymbol{\omega}_*) \ . \tag{2}$$

The main issue with this approach is that $g(\boldsymbol{x}; \boldsymbol{\omega}_*)$ often predicts erratically at out-of-distribution (OOD) inputs. To mitigate this, numerous surrogate or search regularizers have been proposed to either penalize the high-value surrogate prediction at OOD inputs [13, 60, 70] or find an inverse mapping from the desired output to potential inputs [39, 45], as detailed in Section 5. Nonetheless all these approaches are restricted by the limited amount of offline data. Alleviating this bottleneck to reach new SOTA performance is main aim of our proposed approach.

### 2.2 Gaussian Processes

A Gaussian process (GP) [50] defines a probabilistic prior over a random function $h(\boldsymbol{x})$. It is parameterized by a mean function $m(\boldsymbol{x}) = 0^3$ and a kernel function $k(\boldsymbol{x}, \boldsymbol{x}')$. These functions induce a marginal Gaussian prior over the evaluations $\boldsymbol{h} = [h(\boldsymbol{x}_1) \dots h(\boldsymbol{x}_n)]^\top$ of any finite subset of inputs $\{\boldsymbol{x}_1, \dots, \boldsymbol{x}_n\}$. Let $\boldsymbol{x}_\tau$ be an unseen input whose corresponding output $h_\tau = h(\boldsymbol{x}_\tau)$ we wish to predict. The Gaussian prior over $[h(\boldsymbol{x}_1) \dots h(\boldsymbol{x}_n) \ h(\boldsymbol{x}_\tau)]^\top$ implies:

$$h(\boldsymbol{x}_\tau) \mid \boldsymbol{h} \quad \sim \quad \mathbb{N}\big(\boldsymbol{k}_\tau^\top \boldsymbol{K}^{-1} \boldsymbol{h}, k(\boldsymbol{x}_\tau, \boldsymbol{x}_\tau) - \boldsymbol{k}_\tau^\top \boldsymbol{K}^{-1} \boldsymbol{k}_\tau\big) \ , \tag{3}$$

where $\boldsymbol{k}_\tau = [k(\boldsymbol{x}_\tau, \boldsymbol{x}_1) \dots k(\boldsymbol{x}_\tau, \boldsymbol{x}_n)]^\top$ and $\boldsymbol{K}$ denotes the Gram matrix induced on $\{\boldsymbol{x}_1, \dots, \boldsymbol{x}_n\}$ for which $\boldsymbol{K}_{ij} = k(\boldsymbol{x}_i, \boldsymbol{x}_j)$. Assuming a Gaussian likelihood $y \sim \mathbb{N}(h(\boldsymbol{x}), \sigma^2)$, it follows that

$$h(\boldsymbol{x}_\tau) \mid \boldsymbol{y} \quad \sim \quad \mathbb{N}\big(\boldsymbol{k}_\tau^\top (\boldsymbol{K} + \sigma^2 \boldsymbol{I})^{-1} \boldsymbol{y}, k(\boldsymbol{x}_\tau, \boldsymbol{x}_\tau) - \boldsymbol{k}_\tau^\top (\boldsymbol{K} + \sigma^2 \boldsymbol{I})^{-1} \boldsymbol{k}_\tau\big) \ , \tag{4}$$

which explicitly forms the predictive distribution of a Gaussian process. The choice of the kernel function $k(\boldsymbol{x}, \boldsymbol{x}')$ dictates certain properties of the sample functions. In this context, we adopt the commonly used RBF kernel, $k(\boldsymbol{x}, \boldsymbol{x}') = \sigma^2 \exp(-0.5 \cdot \|\boldsymbol{x} - \boldsymbol{x}'\|^2 / \ell^2)$ with the parameter $\sigma^2$ represents the signal variance, controlling the function's amplitude, while $\ell$ denotes the unit length-scale which regulates the function's smoothness. There also exists an extensive literature on improving the complexity of Gaussian process via sparse approximation [47, 48, 56, 41, 24, 42, 57, 8] that enables fast inference with linear complexity in large-scale datasets [24, 30, 31, 28, 32, 29]

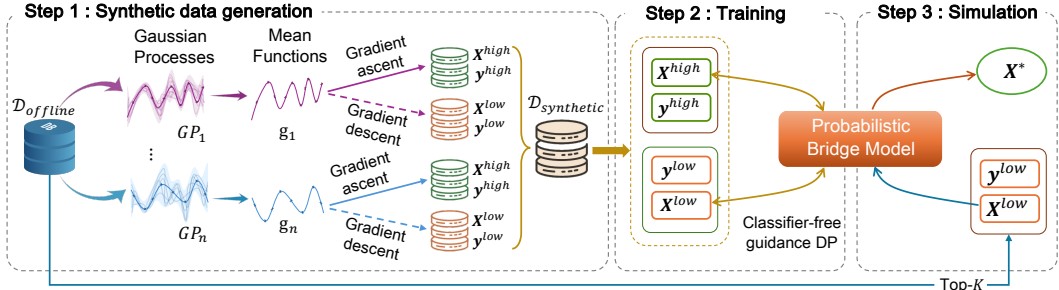

Figure 1: Overview of the **ROOT** workflow: (1) Multiple Gaussian process posteriors are fitted to the offline data, and low- and high-value inputs from the posterior mean functions are sampled to construct a synthetic dataset. (2) This dataset is used to construct our probabilistic bridge model, which learns to map between two different implicit data distributions. (3) The backward process of the learned bridge model is applied to the top-performing inputs from the offline data to generate high-quality candidates for the unknown target function.

## 3 Rethinking Offline Optimization as Distributional Translation

This section introduces a new perspective on offline optimization by framing it as a translation task. We intuitively view the offline data as a source language composed of low-value designs and aim to translate it into a target language of high-value inputs. To enable this translation, we introduce the concept of a probabilistic bridge, which identifies local translation examples by explicitly conditioning on both the source and target contexts. This conditioning enables the construction of feasible transformation paths that connect selected low- and high-value designs within their local neighborhoods, grounding the translation process in meaningful design correspondences (Section 3.1).

These local bridges serve as examples of how to move between regimes when both endpoints are known. The learning algorithm then weaves together these examples to form a global translator capable of generalizing beyond the observed pairs and enabling translation in new contexts where only the source is known. This approach generalizes diffusion models [25], which translate between data and noise, and allows external guiding information to be incorporated, enriching the context and addressing the data bottleneck in offline optimization (Section 3.3).

### 3.1 Probabilistic Bridge for Distributional Translation

This section formalizes the concept of a probabilistic bridge, which generalizes the widely used Denoising Diffusion Probabilistic Model (DDPM) [25]. While DDPM defines a forward diffusion process that maps data to Gaussian noise and learns to reverse it, the probabilistic bridge constructs a space of localized transformation flows conditioned on both the source and the target, representing low-value and high-value designs drawn from implicit data distributions. Learning and applying a probabilistic bridge model for distributional translation involves two phases: a construction phase (Section 3.1) and a learning phase (Section 3.3).

In the construction phase, the transition kernel of each conditioned bridge is derived, capturing a source-to-target translation plan within its local context. These localized examples serve as concrete demonstrations of how to move between regimes. In the learning phase, they are used to train a global, target-agnostic transformation flow that generalizes beyond the observed pairs and enables translation from arbitrary source inputs to valid target outputs. These two phases are described in detail below. A concrete example of a probabilistic bridge is also given in Section 3.2.

#### 3.1.1 Probabilistic Bridge Construction

Given two endpoints $\boldsymbol{x}_0$ and $\boldsymbol{x}_T$, we define a *probabilistic bridge* between them as a discrete observation of a vector-value, time-indexed random function $\boldsymbol{x}_1, \boldsymbol{x}_2, \ldots, \boldsymbol{x}_{T-1}$ which is distributed by Gaussian process [50] with mean function $\psi_t(\boldsymbol{x}_0, \boldsymbol{x}_T)$ and covariance kernel $\kappa_{t,k}\boldsymbol{I}$ [4].

---

[3]We assume a zero mean function since the output can always be re-centered around 0.

[4]We use $\kappa_{t,k}\boldsymbol{I}$ to denote the cross-covariance matrix between $\boldsymbol{x}_t$ and $\boldsymbol{x}_k$ under the joint distribution.

This implies a marginal Gaussian on the probabilistic bridge $x = \text{vec}[x_1, x_2, \ldots, x_{T-1}]$ with mean $\psi = \text{vec}[\psi_1(x_0, x_T); \ldots; \psi_{T-1}(x_0, x_T)]$ which reveals a mass-moving flow between $x_T$ and $x_0$,

$$q(x \mid x_0, x_T) \triangleq \mathbb{N}\Big(x; \psi, \kappa \otimes I\Big) \Rightarrow q(x_t \mid x_0, x_T) = \mathbb{N}\Big(x_t; \psi_t(x_0, x_T), \kappa_{t,t} I\Big), \qquad (5)$$

where $\otimes$ denotes the Kronecker product, $\kappa$ is a matrix containing the entries $\kappa_{t,k}$ and the mean function $\psi_t(x_0, x_T)$ must meet boundary conditions $\psi_0(x_0, x_T) = x_0$ and $\psi_T(x_0, x_T) = x_T$. This further reveals a closed-form expression for backward transition conditioned on both endpoints,

$$q(x_{t-1} \mid x_t, x_0, x_T) = \mathbb{N}\Big(x_{t-1}; \mu(x_t, x_0, x_T), \tilde{\kappa}_{t-1} I\Big), \qquad (6)$$

with the transition mean $\mu(x_t, x_0, x_T) = \psi_{t-1}(x_0, x_T) + \kappa_{t-1,t} \kappa_{t,t}^{-1} (x_t - \psi_t(x_0, x_T))$ and covariance $\tilde{\kappa}_{t-1} = \kappa_{t-1,t-1} - \kappa_{t-1,t} \kappa_{t,t}^{-1} \kappa_{t,t-1}$. This reveals a (step-wise) conditional backward transition that induces a valid transformation flow mapping $x_T$ back to $x_0$. This means for each sampled pair $(x_T, x_0)$, the corresponding target-conditioned backward transition $q(x_{t-1} \mid x_t, x_0, x_T)$ provides an example of a localized transformation flow between them, which can be used to train the desired target-agnostic map $p_\theta(x_{t-1} \mid x_t, x_T)$ as detailed next.

### 3.1.2 Learning Probabilistic Bridge Model

To learn the target-agnostic transformation that maps from a source to a plausible target without knowing it beforehand, we parameterize it as

$$p_\theta(x_{t-1} \mid x_t, x_T) = \mathbb{N}\Big(x_{t-1}; \mu_\theta(x_t, x_T, t), \tilde{\kappa}_{t-1} I\Big). \qquad (7)$$

which uses the same (known) variance parameter as the localized flow/bridge example above and a (learnable) spatio-temporal network $\epsilon_\theta(x_t, t)$ to parameterize the target-agnostic mean transition. This is the centerpiece of the probabilistic bridge model in Eq. 7 which can now be learned via optimizing its parameter $\theta$ to match the example flows generated using Eq. 6,

$$\theta_{\text{PB}} = \arg\min_\theta \mathbb{E}_{(x_0, x_T, t)} \Big[ D_{\text{KL}}\Big( q(x_{t-1} \mid x_t, x_0, x_T) \| p_\theta(x_{t-1} \mid x_t, x_T) \Big) \Big]. \qquad (8)$$

The KL divergence in Eq. 8 above is between two Gaussians and can be computed in closed form which allows for a direct optimization of $\theta$. Given the optimized $\theta_{\text{PB}}$ and a source $x_T$ (low-value design), we can now simulate the generic transformation flow in Eq. 7 to obtain a plausible target $x_0$ (high-value design) which is not known apriori,

$$x_{t-1} = \mu_{\theta_{\text{PB}}}(x_t, x_T, t) + \sqrt{\tilde{\kappa}}\, \epsilon, \qquad (9)$$

where we sample $\epsilon \sim \mathbb{N}(0, I)$ when $t > 1$ and set $\epsilon = 0$ otherwise. This gives us the desired transformation that translates a candidate in the low-value regime to another in the high-value regime. Furthermore, to improve training efficiency, we can also adopt the practical approach in guided diffusion which further conditions $\mu_\theta(x_t, x_T, t) = \mu_\theta(x_t, x_T, y_0, y_T, t)$ on the source and target outputs (i.e., $y_T$ and $y_0$) [26].

### 3.2 Example and Practical Setup

This section provides an working example to substantiate the above probabilistic bridge framework. Choosing $\psi_t(x_0, x_T) = x_0(1 - t/T) + x_T(t/T)$ and $\kappa_{t,k} = (\min(t, k)/T)(1 - \max(t, k)/T)$ results in a Brownian bridge construction,

$$q(x_t \mid x_0, x_T) = \mathbb{N}\left( x_t; x_0\left(1 - \frac{t}{T}\right) + x_T \frac{t}{T}, \frac{t}{T}\left(1 - \frac{t}{T}\right) I \right). \qquad (10)$$

This specifies a transport plan that moves the unit point mass located as $x_T$ to $x_0$. To see this, note that when $t = 0$, Eq. 10 reduces to $\mathbb{N}(x_0, 0)$ and when $t = T$, it reduces to $\mathbb{N}(x_T, 0)$. This setup emphasizes a point-to-point transformation which is particularly suitable for offline optimization.

As we substantiate Eq. 10 with the appropriate (low-value, high-value) pairs $(x_T, x_0)$ and derive examples of localized flow mapping from the low- and high-value regimes following the blueprint in Section 3.1.1, we can learn a target-agnostic transformation consistent with using Eq. 8. This setup

has been used in our experiments to achieve significant performance improvement over prior work, effectively establishing new SOTA. Further details regarding the specific derivation of the Brownian bridge and its practical traning procedure is detailed in Appendix B.2.

**Remark.** The above framework offers a broad and flexible perspective on offline optimization that has not been fully explored in the existing literature. From this viewpoint, there are many promising directions for future investigation. Each valid specification of the mean and kernel functions in the probabilistic bridge model in Eq. 5 defines a distinct way of transporting probability mass from low-value to high-value regions. While offline optimization ultimately requires only one such transport plan, different choices may lead to significantly different learning behaviors and complexities. Understanding how the choice of bridge design influences learning performance remains an important and open question for future work.

▷ Learning $\boldsymbol{\theta}_{\mathrm{PB}}$ however requires access to paired samples of low-value and high-value designs, which is nontrivial in the offline optimization setting where the dataset typically reflects only the low-value regime. This raises a fundamental question: *where can we obtain representative samples from the high-value regime to support bridge construction?* Addressing this challenge is critical for making the probabilistic bridge framework operational, and is the focus of Section 3.3.

### 3.3 Synthetic Data Generation and Practical Black-Box Optimization Algorithm: ROOT

This section revisits the challenge introduced in Section 1: Learning a probabilistic bridge requires examples of localized flows from low- to high-value regimes. Yet, such examples are not available in standard offline datasets, which by design lack high-performing solutions. To address this, we propose a key hypothesis: if a probabilistic bridge can consistently translate low-value inputs to high-value outputs across a sufficiently large set of functions similar to the oracle, it is likely to generalize well to the oracle.

**Motivation.** This motivates an alternative data generation strategy. Rather than depending solely on the offline dataset, we construct a collection of synthetic functions with closed-form structure, allowing us to sample large quantities of low- and high-value inputs. These synthetic examples enable us to train a meta probabilistic bridge with strong zero-shot adaptation. This strategy reflects the common pre-training paradigm in foundation model development, where knowledge accumulated from broad synthetic tasks can be readily transferred to any target task of interest. This motivation leads to our overall workflow, which is illustrated in Figure 1.

**Function Sampling.** Collecting data from such similar functions is fortunately possible using the Gaussian process [50] which specifies a prior over functions around a given mean function. Assuming that the mean function is set to be the oracle, most sampled functions from the corresponding GP will be similar to it. Since we do not know the oracle function, another approach is to begin with a generic GP prior with kernel parameters $\phi = (\sigma, \ell)$ and compute the corresponding GP posterior using the offline data $D_o$. The resulting GP posterior mean is

$$\overline{g}_{\boldsymbol{\phi}_s}(\boldsymbol{x}) \;=\; \boldsymbol{k}(\boldsymbol{\phi}_s)^{\top} \big( \boldsymbol{K}(\boldsymbol{\phi}_s) + \sigma^2 \boldsymbol{I} \big)^{-1} \boldsymbol{y} \,, \tag{11}$$

which has a closed form and is similar the oracle function around the offline data $D_o$. To obtain a wider range of functions similar to the oracle, we compute multiple GP posteriors according to a diverse range of GP priors with different signal and length-scale parameters. These can be obtained from learning a hierarchical GP prior with top-level prior over kernel parameters defining the corresponding GP posteriors at the lower level. We can then sample kernel parameters $\{\phi_s\}_{s=1}^{n_g}$ and extract the corresponding posterior means – see Eq. 11.

**Low- and High-Value Simulation.** Given these functions, we can construct $\boldsymbol{X}_s^{-}$ and $\boldsymbol{X}_s^{+}$ as the set of low- and high-value inputs of $\overline{g}_{\boldsymbol{\phi}_s}(\boldsymbol{x})$ via running $M$-step gradient descent and ascent from a subset of $n_p$ offline input points:

$$\boldsymbol{X}_s^{-} \;\triangleq\; \left\{ \boldsymbol{x}_M^{-} \;\triangleq\; \boldsymbol{x}_0^{-} - \eta \sum_{m=0}^{M-1} \nabla_{\boldsymbol{x}} \overline{g}_{\boldsymbol{\phi}_s}(\boldsymbol{x}_m^{-}) \,\big|_{\boldsymbol{x}_0^{-} \in \mathcal{D}_0} \right\} \,, \tag{12}$$

$$\boldsymbol{X}_s^{+} \;\triangleq\; \left\{ \boldsymbol{x}_M^{+} \;\triangleq\; \boldsymbol{x}_0^{+} + \eta \sum_{m=0}^{M-1} \nabla_{\boldsymbol{x}} \overline{g}_{\boldsymbol{\phi}_s}(\boldsymbol{x}_m^{+}) \,\big|_{\boldsymbol{x}_0^{+} \in \mathcal{D}_0} \right\} \,, \tag{13}$$

where $\boldsymbol{x}_{m+1}^+ \triangleq \boldsymbol{x}_m^+ + \eta \nabla_{\boldsymbol{x}} \overline{g}_{\phi_s}(\boldsymbol{x}_m^+)$ and $\boldsymbol{x}_{m+1}^- \triangleq \boldsymbol{x}_m^- - \eta \nabla_{\boldsymbol{x}} \overline{g}_{\phi_s}(\boldsymbol{x}_m^-)$. Here, $\boldsymbol{x}_0^+ = \boldsymbol{x}_0^- \in D_o$. The corresponding outputs for the above inputs can also be computed using the closed-form of the posterior means. A synthetic dataset $D_s$ collecting low- and high-values from those closed-form posterior mean functions can then be created:

$$D_s = \left\{ (\boldsymbol{X}_s^-, \boldsymbol{y}_s^-), (\boldsymbol{X}_s^+, \boldsymbol{y}_s^+) \right\}_{s=1}^{n_g} \quad \text{where} \quad \boldsymbol{y}_s^- = \overline{g}_{\phi_s}(\boldsymbol{X}_s^-) \quad \text{and} \quad \boldsymbol{y}_s^+ = \overline{g}_{\phi_s}(\boldsymbol{X}_s^+), \quad (14)$$

where $n_g$ is the number of sampled functions.

**Learning Probabilistic Bridge.** Following the blueprint in Section 3.1.1, we can sample $(\boldsymbol{x}_T, y_T, \boldsymbol{x}_0, y_0) \sim D_s$ as training data to train our probabilistic bridge mapping between low- and high-value regions across the aforementioned posterior mean functions, which follows Eq. 8. Furthermore, we also adopt the guided diffusion [26] technique to incorporate the output information $\boldsymbol{y}_T$ and $\boldsymbol{y}_0$ as part of the input to the prediction network $\boldsymbol{\mu}_\theta(\boldsymbol{x}_t, \boldsymbol{x}_T, y_0, y_T, t)$ as stated after Eq. 9.

**Simulation.** Once this probabilistic bridge model has been trained, we can utilize its corresponding (step-wise) source-to-target transition in Eq. 9 to map each offine input (presumbaly in the low-value regime) to a better solution candidate (in a high-value regime). For example, we can run this simulation for the top 128 offline inputs with highest values. The effectiveness of this approach is thoroughly evaluated in Section 4, where we demonstrate its ability to generate high-quality candidates across various benchmarks. A full description of our algorithm and implementation is presented in Appendix B. Our algorithm's computational complexity is discussed in Appendix. C.4.

## 4 Experiments

This section evaluates our proposed method, **ROOT**, through extensive empirical comparisons against various recent baselines on a standard offline optimization benchmark [59]. The experiment setup is summarized in Section 4.1, with detailed results in Section 4.2 and ablation studies in Section 4.3.

### 4.1 Experiments Settings

**Benchmark Tasks.** Our investigation covers four real-world tasks selected from the Design-Bench [59][5] and three RNA-Binding tasks from ViennaRNA [44]. In Design-Bench, the chosen tasks cover both discrete and continuous domains. The discrete tasks, **TF-Bind-8** and **TF-Bind-10** [3], aim to discover DNA sequences with high binding affinity to a specific transcription factor (SIX6 REF R1). On the continuous side, **Ant Morphology** [6] and **D'Kitty Morphology** [2] focus on optimizing the physical structure of a simulated robot ant from OpenAI Gym [6] and the D'Kitty robot from ROBEL [2]. For ViennaRNA, we include three RNA-Binding tasks as RNA-A, RNA-B, and RNA-C [44, 35]. For further details on these tasks, please read Appendix A.

**Baselines.** We selected 21 widely recognized methods, including **BO-qEI** [59], **CMA-ES** [23], **REINFORCE** [67], **COMs** [60], **CbAS** [7], **MINs** [40], **RoMA** [69], **DDOM** [39], **ICT** [70], **Tri-mentoring** [13], **GTG** [72], **BDI** [12], **RGD** [11], **LTR** [55], **BONET** [38], **MATCH-OPT** [27], **PGS** [10], **GABO** [68], **DEMO** [71], **GA on GP** (Section 4.3) and standard **GA**.

**Evaluation Protocol.** Following the approach in [59], each method generates 128 optimized design candidates, which are then evaluated by the oracle function. The performances are ranked, and results are recorded at the 50th, 80th, and 100th percentiles. To ensure consistency, all results are averaged over 8 independent runs with reported standard deviation.

**Hyper-parameter Configuration.** For each baseline, we adopt the optimized settings from the original papers. For GP kernel hyper-parameters in our data generation, we sample lengthscales $\ell_s$ and variances $\sigma_s^2$ uniformly from $[\ell_0 - \delta, \ell_0 + \delta]$ and $[\sigma_0^2 - \delta, \sigma_0^2 + \delta]$, with $\ell_0 = \sigma_0^2 = 1.0$ for continuous tasks and 6.25 for discrete tasks, and $\delta = 0.25$. We use $M = 100$ gradient steps with step sizes 0.001 (continuous) and 0.05 (discrete). Additional data generation details are in Appendix B.1 and Table 6. For training the Probabilistic Bridge model, we use a Brownian Bridge diffusion process with the Adam optimizer over $E = 100$ epochs and $n_g = 800$ synthetic functions, running on a single NVIDIA A100-SXM4-80GB GPU. More training details are provided in Appendix B.2.

---

[5]We exclude tasks marked by previous works as having high oracle function noise and inaccuracy (**ChEMBL**, **Hopper**, and **Superconductor**) and tasks that require substantial computational resources to evaluate (**NAS**)

## 4.2 Experimental Results

Table 1: Experiment results on Design-Bench Tasks. We report the maximum score ($100^{th}$ percentile) among $Q = 128$ candidates. Blue denotes the best entry in the column, while Brown indicates the second best. **Mean Rank** is the average rank across all benchmark tasks.

| | | Benchmarks | | | | |
| Method | Ant | D'Kitty | TFBind8 | TFBind10 | Mean Rank |
|---|---|---|---|---|---|
| $D_o$ (best) | 0.565 | 0.884 | 0.439 | 0.467 | - |
| BO-qEI | 0.812 ± 0.000 | 0.896 ± 0.000 | 0.825 ± 0.091 | 0.627 ± 0.033 | 16.75 / 22 |
| CMA-ES | 1.561 ± 0.896 | 0.724 ± 0.001 | 0.939 ± 0.039 | 0.664 ± 0.034 | 8.00 / 22 |
| REINFORCE | 0.263 ± 0.026 | 0.573 ± 0.204 | 0.961 ± 0.034 | 0.618 ± 0.011 | 17.00 / 22 |
| GA | 0.293 ± 0.029 | 0.860 ± 0.021 | 0.985 ± 0.011 | 0.638 ± 0.032 | 12.75 / 22 |
| COMs | 0.882 ± 0.044 | 0.932 ± 0.006 | 0.940 ± 0.027 | 0.621 ± 0.033 | 13.25 / 22 |
| CbAS | 0.846 ± 0.033 | 0.895 ± 0.016 | 0.903 ± 0.028 | 0.649 ± 0.055 | 12.50 / 22 |
| MINs | 0.894 ± 0.022 | 0.939 ± 0.004 | 0.908 ± 0.063 | 0.630 ± 0.019 | 12.50 / 22 |
| GA on GP | 0.948 ± 0.013 | 0.946 ± 0.001 | 0.770 ± 0.087 | 0.654 ± 0.038 | 9.25 / 22 |
| RoMA | 0.593 ± 0.066 | 0.829 ± 0.020 | 0.665 ± 0.000 | 0.553 ± 0.000 | 20.00 / 22 |
| ICT | 0.911 ± 0.030 | 0.945 ± 0.011 | 0.888 ± 0.047 | 0.624 ± 0.033 | 13.50 / 22 |
| Tri-mentoring | 0.944 ± 0.033 | 0.950 ± 0.015 | 0.899 ± 0.045 | 0.647 ± 0.039 | 9.00 / 22 |
| MATCH-OPT | 0.931 ± 0.011 | 0.957 ± 0.014 | 0.977 ± 0.004 | 0.543 ± 0.002 | 9.50 / 22 |
| PGS | 0.949 ± 0.017 | 0.966 ± 0.013 | 0.981 ± 0.015 | 0.532 ± 0.000 | 7.75 / 22 |
| LTR | 0.907 ± 0.032 | 0.960 ± 0.014 | 0.973 ± 0.000 | 0.652 ± 0.039 | 6.25 / 22 |
| DDOM | 0.930 ± 0.029 | 0.925 ± 0.008 | 0.885 ± 0.061 | 0.634 ± 0.015 | 13.75 / 22 |
| GTG | 0.865 ± 0.040 | 0.935 ± 0.010 | 0.901 ± 0.039 | 0.639 ± 0.016 | 12.50 / 22 |
| BDI | 0.964 ± 0.000 | 0.941 ± 0.000 | 0.973 ± 0.000 | 0.636 ± 0.020 | 7.50 / 22 |
| RGD | 0.922 ± 0.020 | 0.883 ± 0.014 | 0.889 ± 0.068 | 0.644 ± 0.048 | 13.00 / 22 |
| BONET | 0.948 ± 0.025 | 0.957 ± 0.008 | 0.894 ± 0.086 | 0.606 ± 0.024 | 10.75 / 22 |
| GABO | 0.224 ± 0.051 | 0.719 ± 0.001 | 0.939 ± 0.038 | 0.639 ± 0.033 | 15.25 / 22 |
| DEMO | 0.948 ± 0.013 | 0.956 ± 0.011 | 0.812 ± 0.054 | 0.648 ± 0.042 | 9.25 / 22 |
| **ROOT (ours)** | 0.965 ± 0.014 | 0.972 ± 0.005 | 0.986 ± 0.007 | 0.685 ± 0.053 | 1.25 / 22 |

Table 2: Experiments on Biological RNA Design Tasks. We report the maximum score among $Q = 128$ candidates.

| | | Benchmarks | | | |
| Method | RNA-A | RNA-B | RNA-C | Mean Rank |
|---|---|---|---|---|
| CbAS | 0.270 ± 0.098 | 0.249 ± 0.088 | 0.261 ± 0.093 | 6.00 / 8 |
| BO-qEI | 0.537 ± 0.106 | 0.517 ± 0.108 | 0.481 ± 0.100 | 3.67 / 8 |
| GA | 0.518 ± 0.120 | 0.499 ± 0.100 | 0.496 ± 0.091 | 4.33 / 8 |
| COMs | 0.187 ± 0.123 | 0.144 ± 0.121 | 0.209 ± 0.100 | 7.67 / 8 |
| REINFORCE | 0.166 ± 0.096 | 0.149 ± 0.081 | 0.225 ± 0.075 | 7.33 / 8 |
| BDI | 0.604 ± 0.000 | 0.505 ± 0.000 | 0.411 ± 0.000 | 4.00 / 8 |
| Boot-Gen | 0.913 ± 0.064 | 0.881 ± 0.024 | 0.786 ± 0.039 | 2.00 / 8 |
| **ROOT (ours)** | 0.956 ± 0.023 | 0.955 ± 0.013 | 0.922 ± 0.013 | 1.00 / 8 |

Table 3: Effect of initial-point selection on **ROOT**'s performance.

| Type | Ant | TFBind8 |
|---|---|---|
| Random | 0.953 ± 0.014 | 0.976 ± 0.007 |
| Lowest | 0.545 ± 0.214 | 0.969 ± 0.009 |
| Highest | 0.965 ± 0.014 | 0.986 ± 0.007 |

Table 4: **ROOT** vs ExPT in few-shot settings.

| Method | Ant | TFBind8 |
|---|---|---|
| ExPT | 0.940 ± 0.027 | 0.874 ± 0.071 |
| **ROOT** | 0.942 ± 0.035 | 0.895 ± 0.086 |

This section compares our method to 21 baselines, evaluating the 50th, 80th, and 100th percentiles. Due to limited space, we report only the 100th percentile results in the main text; details on the 50th and 80th percentiles and score distributions for our method versus others are in Appendix C.2.

**Results on Continuous Tasks:** Table 1 (first two columns) shows our continuous-task results. On D'Kitty, we set a new SOTA at **0.972** with a small standard deviation **0.005**. On Ant, although CMA-ES surpasses us at the 100th percentile, its standard deviation (**0.896**) is nearly **64×** ours (**0.014**) and it collapses near zero at the 80th/50th percentiles (see Appendix C.2).

**Results on Discrete Tasks:** Table 1 (last two columns) present our discrete-task results. Our **ROOT** achieves both the top rank for the TFBind10 task with a mean score **0.685** and the TFBind8 task with **0.986**. Our standard deviation **0.007** for the TFBind8 task is even smaller than that of all other baselines, except for RoMa, which performs poorly.

**Results on RNA-Binding Tasks**: Table 2 demonstrates **ROOT**'s superior performance, ranking first on all three RNA benchmarks. We outperform Boot-Gen[35] by margins of **0.043** (RNA-A), **0.074**

Table 5: Performance on **TF-Bind-8** and **Ant** of different methods trained by only the $p\%$ poorest-performing designs from the offline dataset.

| Method | TF-Bind-8 | | | Ant | | |
|---|---|---|---|---|---|---|
| | 50% | 20% | 10% | 50% | 20% | 10% |
| GA | $0.580 \pm 0.199$ | $0.480 \pm 0.218$ | $0.559 \pm 0.170$ | $0.394 \pm 0.023$ | $0.663 \pm 0.065$ | $0.619 \pm 0.120$ |
| COMs | $0.935 \pm 0.052$ | $0.872 \pm 0.085$ | $0.771 \pm 0.128$ | $0.898 \pm 0.035$ | $0.880 \pm 0.027$ | $0.845 \pm 0.041$ |
| REINFORCE | $0.915 \pm 0.039$ | $0.917 \pm 0.040$ | $0.913 \pm 0.038$ | $0.317 \pm 0.016$ | $0.261 \pm 0.052$ | $0.281 \pm 0.034$ |
| LTR | $0.959 \pm 0.022$ | $0.927 \pm 0.033$ | $0.909 \pm 0.034$ | $0.909 \pm 0.042$ | $0.871 \pm 0.059$ | $0.813 \pm 0.026$ |
| **ROOT** | $0.964 \pm 0.015$ | $0.946 \pm 0.045$ | $0.915 \pm 0.019$ | $0.909 \pm 0.012$ | $0.930 \pm 0.023$ | $0.861 \pm 0.051$ |

(RNA-B), and **0.136** (RNA-C), while our standard deviations are roughly half those of Boot-Gen, achieving better performance and lower variance (hence, more stable).

Overall, **ROOT** achieved a mean rank of **1.25** across both discrete and continuous domains, setting a new SOTA on 3 out of 4 Design-Bench tasks, as well as on all three RNA-binding tasks. This demonstrates the robust and consistent effectiveness of **ROOT** across diverse settings.

## 4.3 Ablation Experiments

**Selection Strategies for Initial Points.** We explore 3 initial point strategies for $\boldsymbol{x}_0^+, \boldsymbol{x}_0^-$ in (Eq. 12 and Eq. 13): random sampling, and selecting points with the lowest or highest objective values from $D_o$. As shown in Table 3, the third strategy achieves the best results.

**Few-Shot Experimental Designs Setting.** In offline optimization, the few-shot experimental designs (ED) setting, introduced in ExPT [45], presents a more challenging task where only a small set of labeled data points, $D_{\text{label}} = \{(\boldsymbol{x}_i, y_i)\}_{i=1}^{n_l}$ is available alongside a larger set of unlabeled data, $D_{\text{unlabeled}} = \{\boldsymbol{x}_i\}_{i=1}^{n_u}$. To evaluate our model in this scenario, we follow ExPT's protocol, using a random 1% of the offline data as labeled points and the remaining 99% as unlabeled. For synthetic function generation, we first fit a prior Gaussian process (GP) to $D_{\text{label}}$, then use the posterior to generate pseudo-labels for $D_{\text{unlabeled}}$. We combine the labeled and pseudo-labeled data and fit another GP to this dataset. The mean function of this refitted GP is used as the synthetic function, following the same procedure as in the main method. As shown in Table 4, **ROOT** outperforms the ExPT model in this setting by substantial margins.

**Poor Offline data Coverage Setting.** Beyond the Few-Shot Experimental Design scenario with limited coverage, we further evaluated **ROOT**'s robustness under varying dataset support quality. Specifically, we trained **ROOT** and the baselines using only the $p\%$ lowest-performing designs from the offline dataset. As shown in Table 5, **ROOT** achieves the best performance across all settings on both **TF-Bind-8** and **Ant**. These results demonstrate that **ROOT** adapts effectively in challenging scenarios where training data is heavily biased toward low-quality designs.

**Effectiveness of Learning the Probabilistic Bridge Model.** To demonstrate/ablate the effectiveness of our learning probabilistic bridge model (see Section 3.1.1), we further conducted an experiment that runs gradient ascent on GP posterior mean function for comparison. This simple baseline is denoted as **GA on GP** in Table 1. It is observed that our method significantly outperforms this baseline, confirming the impact of learning probabilistic bridge.

**Number of Gradient Steps ($M$).** We experimented with various numbers of gradient steps ($M$), from the set $\{25, 50, 75, 100\}$ to construct $\boldsymbol{X}_s^-$ and $\boldsymbol{X}_s^+$ during the data generation phase (see Section 3.3). Our experiments reveal that increasing $M$ consistently improves the overall performance of the algorithm as illustrated in Table 6. Increasing the number of gradient steps allows our model to more precisely distinguish between low-value and high-value regions in the distribution, substantially enhancing our performance. However, increasing the number of gradient steps also increases computational time, so we select $M = 100$ as the best balance between performance and efficiency.

**Number of Initial ($n_p$).** For each synthetic function generated by the Gaussian process, we will draw a number of initial data points ($n_p$) from the offline dataset to initiate the exploration into the low-value and high-value regions via gradient descent and ascent, respectively. We experimented with different numbers of initial points from the set $\{128, 256, 512, 1024\}$. As shown in Table 7, increasing the number of initial points $n_p$ consistently improves performance. This observation is similar to a previous observation that having more well-curated training data tends to enhance the overall performance. We selected $n_p = 1024$ as the best balance between cost and performance.

Table 6: Impact of gradient steps $M$ on **ROOT**.

| Steps ($M$) | Ant | D'Kitty | TF-Bind-8 | TF-Bind-10 |
|---|---|---|---|---|
| 25 | 0.968 ± 0.009 | 0.972 ± 0.003 | 0.952 ± 0.024 | 0.640 ± 0.039 |
| 50 | 0.968 ± 0.015 | 0.969 ± 0.003 | 0.945 ± 0.025 | 0.639 ± 0.024 |
| 75 | 0.950 ± 0.011 | 0.969 ± 0.005 | 0.972 ± 0.013 | 0.652 ± 0.033 |
| 100 | 0.965 ± 0.014 | 0.972 ± 0.005 | 0.986 ± 0.007 | 0.685 ± 0.053 |

Table 7: Impact of number of initial data points on **ROOT**.

| Initial Points ($n_p$) | Ant | TF-Bind-8 |
|---|---|---|
| 128 | 0.915 ± 0.020 | 0.948 ± 0.024 |
| 256 | 0.950 ± 0.010 | 0.968 ± 0.018 |
| 512 | 0.964 ± 0.010 | 0.974 ± 0.006 |
| 1024 | 0.965 ± 0.014 | 0.986 ± 0.007 |

**Additional Ablation Studies**. We further performed a series of ablation studies, examining key hyperparameters of our method, alternative strategies for sampling GP kernel parameters, different data sampling techniques beyond GP, the impact of measurement noise, performance on high-dimensional continuous tasks, and settings with limited query budgets. For completeness, we also report computational complexity analysis and empirical runtime comparisons of **ROOT** against baselines. Due to space constraints, all these results are deferred to Appendices C.

## 5 Related Works

Existing approaches in offline optimization can be categorized into three main families: **forward modeling**, **inverse modeling**, and **learning search policies**.

**Forward Modeling** tackles out-of-distribution (OOD) issues by penalizing high surrogate predictions on OOD inputs [60, 13, 70, 69, 14, 15, 27, 46, 19, 18]. For example, COM [60] identifies OOD regions early during gradient updates and re-trains the surrogate with regularizers to penalize high-value predictions at these inputs. BOSS[14] introduces a sensitivity-aware regularizer for offline optimizers, while ICT, Tri-mentoring [70, 13] use co-teaching among surrogates to improve performance.

**Inverse Modeling** avoids OOD problems by directly learning high-value regions [40, 45, 39]. For instance, MIN [40] uses model inversion networks to map scores back to inputs, while ExPT [45] combines unsupervised learning and few-shot experimental design for optimizing synthetic functions. DDOM [39] develops a guided diffusion model to generate designs conditioned on function values. The model is trained using weighted sampling from the offline dataset.

**Learning Search Policies** aims to replicate optimization paths from low- to high-value designs [38, 10]. BONET [38] synthesizes trajectories from offline data using a heuristic for monotonic transitions and trains an auto-regressive model. PGS [10] reinterprets offline optimization as a reinforcement learning task, which optimizes for an effective policy using sampled trajectories from offline data.

Overall, these methods remain constrained by the availability of offline data. For instance, DDOM [39] employs guided diffusion to learn an inverse mapping from desired performance outputs to potential input designs. However, the adopted diffusion model must be trained on weighted sampling from the offline data, which may lack critical information regarding potential high-performing input regions that are far from the offline regimes.

To address the challenges posed by limited data, we reframe offline optimization as a distributional translation task. This perspective unveils an intriguing direction: rather than depending solely on scarce high-value observations, one can learn a global translation model by stitching together localized transformation examples between low- and high-value regimes. This allows the optimization process to be guided by a learned probabilistic bridge that generalizes across design landscapes, offering a flexible and data-efficient alternative to traditional surrogate-based methods.

## 6 Conclusion

We proposed a new perspective on offline black-box optimization by reframing it as a distributional translation task between low-value and high-value input regimes. At the core of this approach is the *probabilistic bridge*, a model that learns localized transformation flows conditioned on both source and target designs, and synthesizes them into a global translation mechanism. To address the lack of high-value examples in offline datasets, we introduced a synthetic data generation framework that enables pre-training of a meta probabilistic bridge with strong zero-shot generalization. This shifts the focus from modeling the objective function to modeling design transitions, opening new possibilities for data-efficient optimization. Future directions include exploring alternative bridge parameterizations and extending the framework to more complex optimization settings.

## Acknowledgement

This work utilized GPU compute resource at SDSC and ACES through allocation CIS230391 from the Advanced Cyberinfrastructure Coordination Ecosystem: Services and Support (ACCESS) program [4], which is supported by U.S. National Science Foundation grants #2138259, #2138286, #2138307, #2137603, and #2138296.

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

# A  Task Details

## A.1  Design-Bench Task

Design-Bench [59] is a widely adopted benchmark for evaluating offline black-box optimization algorithms. Table 8 presents the summary of five evaluation tasks in Design-Bench.

Table 8: Overview of tasks in the Design-Bench benchmark. The ground-truth oracle functions for these tasks (unknown to our algorithm) are accessible during evaluation.

| Task | Offline Size | Input Dimension | Input Category | Input Type | Oracle |
|------|------|------|------|------|------|
| TF Bind 8 | 32,898 | 8 | 4 | Discrete | Exact |
| TF Bind 10 | 10,000 | 10 | 4 | Discrete | Exact |
| Ant Morphology | 10,004 | 60 | N/A | Continuous | Exact |
| D'Kitty Morphology | 10,004 | 56 | N/A | Continuous | Exact |
| RNA-Binding | 5000 | 14 | 4 | Discrete | Exact |

**TF Bind 8 and TF Bind 10: DNA Sequence Optimization.** The goals of the **TF Bind 8** and **TF Bind 10** tasks are to discover, respectively, the length-8 and length-10 DNA sequences with the strongest binding affinity to a specific transcription factor (`SIX6_REF_R1` by default) [3]. In these settings, each DNA candidate is a sequence of nucleotides where each nucleotide has 4 possible categorical values. The Design-Bench benchmark grants access to the full oracle functions for the **TF-Bind-8** and **TF-Bind-10** tasks, which correspond to databases containing exact binding affinities for all possible sequences (i.e., $4^8$ and $4^{10}$ combinations, respectively). Following the evaluation protocol in [59], a fixed subset of sequences is sampled from each oracle and used as the offline dataset for all baselines. In particular, **TF-Bind-8** provides an offline dataset of $32,898$ sequences while **TF-Bind-10** provides an offline dataset with $10,000$ sequences.

**Ant and D'Kitty Morphology: Robot Morphology Optimization.** These tasks focus on optimizing the physical structure of two simulated robots: (1) Ant from OpenAI Gym [6] and (2) D'Kitty from ROBEL [2]. In **Ant Morphology**, the objective is to find the structure of a quadruped robot to maximize its running speed. In **D'Kitty Morphology** the goal is to find the most effective structure for the D'Kitty robot that enables it to reach a specific target. In particular, each structure candidate specifies the morphology parameters, such as the size, orientation, and placement of limbs, for a robot controller which will be trained using the Soft Actor-Critic algorithm [22]. For the Ant robots, there are 60 morphology parameters. For the D'Kitty robots, there are 56 morphology parameters. The solution quality of each structure candidate is obtained by simulating the corresponding trained controller in the MuJoCo [58] environment. Each simulation runs for 100 steps and the overall solution quality is obtained by averaging simulation results across 16 independent runs.

## A.2  RNA Task

**RNA-Binding** [44]. This is an inverse-folding task whose objective is to optimize RNA sequences of fixed length (14 nucleotides) over the alphabet $\{A, U, C, G\}$, which are abbreviations for adenine, uracil, cytosine, and guanine, respectively. In particular, the task is to find a sequence whose predicted minimum-free-energy (MFE) structure maximizes its binding affinity to a given transcription factor. We follow the benchmark setup of Bootgen [35] by considering three target structures **RNA-A** (L14 RNA1), **RNA-B** (L14 RNA2), and **RNA-C** (L14 RNA3). To construct the offline dataset $D_o$, we use the FLEXS codebase [51] to generate sequences uniformly at random. The RNAinverse algorithm from ViennaRNA 2.0 is then applied iteratively until 5,000 sequences with minimum free energy (MFE) below 0.12 are obtained, forming the final offline dataset.

# B  Detailed Algorithmic Description and Implementation of ROOT

## B.1  Generating Synthetic Data with GP

To generate a diverse range of synthetic functions which are sufficiently similar to the oracle, we first sample the kernel parameters $\ell_s$ (lengthscale) and $\sigma_s^2$ (variance) uniformly from the ranges $[\ell_0 - \delta, \ell_0 + \delta]$ and $]\sigma_0^2 - \delta, \sigma_0^2 + \delta]$, where $\ell_0$, $\sigma_0^2$ and $\delta$ are fixed initial hyperparameters, as reported

Table 9: Hyperparameters for the synthetic data generation of **ROOT**.

| Hyperparameter | Value |
|---|---|
| $\ell_0, \sigma_0^2$ | 1.0 (continuous) 6.25 (discrete) |
| $\delta$ | 0.25 |
| Step size ($\eta$) | 0.001 (continuous) 0.05 (discrete) |
| Number of gradient steps ($M$) | 100 |
| Threshold ($\tau$) | 0.001 |

---

**Algorithm 1** Synthetic Data Generation via Simulating Gaussian Process (GP) Posteriors

---

1: **Input:** Offline dataset $D_o = \{x_i, y_i\}_{i=1}^n$, number of functions per epoch $n_e$, number of points per function $n_p$, number of gradient steps $M$
2: **Output:** Synthetic dataset $D_s = \{(X^-, y^-), (X^+, y^+)\}$
3: $\mathcal{D}_s \leftarrow \emptyset$
4: **for** $s = 1$ **to** $n_e$ **do**
5:      Sample kernel parameters $\phi_s = (\ell_s, \sigma_s^2)$: $\ell_s \sim U(\ell_0 - \delta, \ell_0 + \delta)$, $\sigma_s^2 \sim U(\sigma_0^2 - \delta, \sigma_0^2 + \delta)$
6:      Compute the mean function $\bar{g}_{\phi_s}$ of the posterior Gaussian process via Eq. 11
7:      Sample a subset of $n_p$ points from offline data: $\mathcal{D}_0 \subset D_o$ where $|\mathcal{D}_0| = n_p$
8:      Compute the set of low-value designs $X_s^-$ using Eq. 12
9:      Compute the set of high-value designs $X_s^+$ using Eq. 13
10:      Compute corresponding low and high scores: $y_s^- = \bar{g}_{\phi_s}(X_s^-)$, $y_s^+ = \bar{g}_{\phi_s}(X_s^+)$
11:      $D_s \leftarrow D_s \cup \{(X_s^-, y_s^-), (X_s^+, y_s^+)\}$
12: **end for**
13: **Return** $D_s$

---

in Table 9. After sampling, we compute the mean function of the Gaussian process posterior based on the offline data. We then sample $n_p$ points from the offline data and perform $M = 100$ gradient ascent and gradient descent steps with a step size $\eta$ (more details are provided in Section 3.3). To enhance the quality of our synthetic data, we filter out any pair of low- and high-value points $(x^-, y^-)$ and $(x^+, y^+)$ whose difference between $y^+$ and $y^-$ is smaller than a threshold $\tau$. All key hyperparameters for this process are listed in Table 9. The pseudocode of the algorithm is presented in Algorithm 1.

### B.2 Learning the Probabilistic Bridge Model

This section provides further details regarding the implementation of the learning loss in Eq. 8 with respect to its Brownian bridge instantiation in Section 3.2. In particular, following the formulation of $q(x_t \mid x_0, x_T)$ in Eq. 10, we can express $x_t$

$$x_t = \left(1 - \frac{t}{T}\right) x_0 + \frac{t}{T} x_T + \sqrt{\kappa_{t,t}} \epsilon_t \tag{15}$$

where $\epsilon_t \sim \mathbb{N}(0, I)$ and $\kappa_{t,t} = 2(m_t - m_t^2)$ with $m_t = t/T$. This reproduces the Brownian Bridge Diffusion Model [43]. Following the derivation in [43], we obtain:

$$q(x_{t-1} \mid x_t, x_0, x_T) = \mathbb{N}\left(\mu(x_t, x_0, x_T), \tilde{\kappa}_{t-1} \cdot I\right),$$

where the mean is

$$\mu(x_t, x_0, x_T) = u_t \cdot x_t + v_t \cdot x_T + w_t \cdot \left(m_t(x_T - x_0) + \sqrt{\kappa_{t,t}} \epsilon\right), \tag{16}$$

and the variance is

$$\tilde{\kappa}_{t-1} = \left(\kappa_{t,t} - \kappa_{t-1,t-1} \cdot \frac{(1 - m_t)^2}{(1 - m_{t-1})^2}\right) \cdot \frac{\kappa_{t-1,t-1}}{\kappa_{t,t}}. \tag{17}$$

**Algorithm 2** Learning the Probabilistic Bridge and Simulation for Offline Optimization (**ROOT**)

---

1: **Input:** Offline dataset $D_o = \{\boldsymbol{x}_i, y_i\}_{i=1}^n$, number of epochs $E$, number of diffusion steps $T$, scale $\alpha$, best objective value $y_*$, conditional dropout probability $\rho$, number of iterations $I$
2: **Output:** High-value candidate $\boldsymbol{x}_0^*$
3:
4: **Learning Probabilistic Bridge Phase:**
5: Initialize model parameters $\boldsymbol{\theta}$
6: **for** $e = 1$ **to** $E$ **do**
7:     Generate synthetic dataset $D_s$ from Algorithm 1
8:     **for** $i = 1$ **to** $I$ **do**
9:         Sample $\{\boldsymbol{x}_T, y_T, \boldsymbol{x}_0, y_0\} \sim \mathcal{D}_s = \{\boldsymbol{X}^-, \boldsymbol{y}^-, \boldsymbol{X}^+, \boldsymbol{y}^+\}$
10:         Sample timestep $t \sim \text{Uniform}(1, T)$, $\gamma \sim \text{Ber}(\rho)$, $y = [y_T, y_0]$
11:         Forward diffusion: $\boldsymbol{x}_t = (1 - m_t) \cdot \boldsymbol{x}_0 + m_t \cdot \boldsymbol{x}_T + \sqrt{\kappa_{t,t}}\boldsymbol{\epsilon}$ where $\boldsymbol{\epsilon} \sim \mathbb{N}(0, \boldsymbol{I})$
12:         Gradient descent step: $\nabla_\theta \left\| m_t(\boldsymbol{x}_T - \boldsymbol{x}_0) + \sqrt{\kappa_{t,t}}\boldsymbol{\epsilon} - \boldsymbol{\epsilon}_\theta(\boldsymbol{x}_t, t, (1 - \gamma) \cdot y + \gamma \cdot \emptyset) \right\|^2$ (Eq. 24)
13:     **end for**
14: **end for**
15:
16: **Simulation Phase:**
17: $\{\boldsymbol{x}_T, y_T\} \leftarrow$ 128 best designs in $D_o$, $y = [y_T, \alpha \cdot y_*]$
18: **for** $t = T$ **to** $1$ **do**
19:     $\boldsymbol{z} \sim \mathbb{N}(0, \boldsymbol{I})$ if $t > 0$ else $\boldsymbol{z} = 0$
20:     Compute $\boldsymbol{\epsilon}_\theta(\boldsymbol{x}_t, t, y)$ via Eq. 25
21:     $\boldsymbol{x}_{t-1} = u_t \cdot \boldsymbol{x}_t + v_t \cdot \boldsymbol{x}_T + w_t \cdot \boldsymbol{\epsilon}_\theta(\boldsymbol{x}_t, t, y) + \sqrt{\tilde{\kappa}_{t-1}} \cdot \boldsymbol{z}$
22: **end for**
23:
24: **Return:** high-value design $\boldsymbol{x}_0^* = \boldsymbol{x}_0$

---

Furthermore, the coefficients for the above mean function were derived as

$$u_t = \frac{\kappa_{t-1,t-1}}{\kappa_{t,t}} \cdot \frac{1 - m_t}{1 - m_{t-1}} + \frac{\kappa_{t,t} - \kappa_{t-1,t-1} \cdot \frac{(1-m_t)^2}{(1-m_{t-1})^2}}{\kappa_{t,t}} \cdot (1 - m_{t-1}), \quad (18)$$

$$v_t = m_{t-1} - m_t \cdot \frac{1 - m_t}{1 - m_{t-1}} \cdot \frac{\kappa_{t-1,t-1}}{\kappa_{t,t}}, \quad (19)$$

$$w_t = (1 - m_{t-1}) \cdot \frac{\kappa_{t,t} - \kappa_{t-1,t-1} \cdot \frac{(1-m_t)^2}{(1-m_{t-1})^2}}{\kappa_{t,t}}. \quad (20)$$

This setup thus provides training examples of localized flows which can be used to learn a parameterized target-agnostic transformation that maps from a source $\boldsymbol{x}_T$ to a plausible target $\boldsymbol{x}_0$ without knowing it beforehand using the loss in Eq. 8. To implement this loss, we parameterize the transition probability of the aforementioned target-agnostic transformation $p_\theta(\boldsymbol{x}_{t-1}|\boldsymbol{x}_t, \boldsymbol{x}_T)$ as:

$$p_\theta(\boldsymbol{x}_{t-1}|\boldsymbol{x}_t, \boldsymbol{x}_T) = \mathbb{N}\Big(\boldsymbol{x}_{t-1}; \boldsymbol{\mu}_\theta(\boldsymbol{x}_t, \boldsymbol{x}_T, t), \tilde{\kappa}_{t-1} \cdot \boldsymbol{I}\Big), \quad (21)$$

where the $\boldsymbol{\mu}_\theta$ is parameterized following Eq. 16:

$$\boldsymbol{\mu}_\theta(\boldsymbol{x}_t, \boldsymbol{x}_T, t) = u_t \cdot \boldsymbol{x}_t + v_t \cdot \boldsymbol{x}_T + w_t \cdot \boldsymbol{\epsilon}_\theta(\boldsymbol{x}_t, t), \quad (22)$$

which features a parameterized noise prediction network $\boldsymbol{\epsilon}(\boldsymbol{x}_t, t)$ to predict the noise perturbed quantity $m_t(\boldsymbol{x}_T - \boldsymbol{x}_0) + \sqrt{\kappa_{t,t}}\boldsymbol{\epsilon}$ of a local flow at time $t$. Under this parameterization, the training loss in Eq. 8 can be simplified as,

$$\boldsymbol{\theta}_{\text{PB}} = \arg\min_{\boldsymbol{\theta}} \mathbb{E}_{\boldsymbol{x}_0, \boldsymbol{x}_T, \boldsymbol{\epsilon}}\Big[\|m_t(\boldsymbol{x}_T - \boldsymbol{x}_0) + \sqrt{\kappa_{t,t}}\boldsymbol{\epsilon} - \boldsymbol{\epsilon}_\theta(\boldsymbol{x}_t, t)\|^2\Big]. \quad (23)$$

It is then approximately optimized via sampling $(\boldsymbol{x}_T, \boldsymbol{x}_0)$ from the synthetic dataset $D_s$ created using Algorithm 1 above. For a more practical implementation, we further leverage the corresponding socres $y_T$ and $y_0$ of $\boldsymbol{x}_T$ and $\boldsymbol{x}_0$ which are also included in $D_s$. In particular, we further parameterize $\boldsymbol{\epsilon}_\theta(\boldsymbol{x}_t, t)$ as $\boldsymbol{\epsilon}_\theta(\boldsymbol{x}_t, t, (1 - \gamma)y + \gamma\emptyset)$ with $y = (y_0, y_T)$ and $\gamma \sim \text{Ber}(\rho)$ following similar practice

in guided diffusion [26]. The intuition is that we want to leverage the output scores to guide the noise prediction network but ath the same time, we do not want the network to overly depend on such guidance. This is enforced using the dropout trick which randomly removes the guiding information during training with probability $\rho \in (0, 1)$. The above learning loss can then be recast as:

$$\boldsymbol{\theta}^*_{\text{PB}} \triangleq \underset{\boldsymbol{\theta}}{\arg\min} \; \underset{\boldsymbol{x}_0, y_0, \boldsymbol{x}_T, y_T, \boldsymbol{\epsilon}}{\mathbb{E}} \left[ \| m_t(\boldsymbol{x}_T - \boldsymbol{x}_0) + \sqrt{\kappa_{t,t}}\boldsymbol{\epsilon} - \boldsymbol{\epsilon}_\theta(\boldsymbol{x}_t, t, (1 - \gamma) \cdot y + \gamma \cdot \emptyset) \|^2 \right] , \quad (24)$$

where $\gamma \sim \text{Ber}(\rho)$. Once trained, the noise network $\boldsymbol{\epsilon}_\theta$ is used as below during the simulation phase:

$$\boldsymbol{\epsilon}_\theta(\boldsymbol{x}_t, t, y) = (1 + \beta) \cdot \boldsymbol{\epsilon}_\theta(\boldsymbol{x}_t, t, y) - \beta \cdot \boldsymbol{\epsilon}_\theta(\boldsymbol{x}_t, t, \emptyset) , \quad (25)$$

where $\beta$ is the classifier free guidance weight and $y = [y_T, \alpha \cdot y_*]$ with $y_*$ denotes the maximum oracle score. Our detail algorithm is presented in Algorithm 2.

For more details, we utilize an MLP $\boldsymbol{\epsilon}_\theta(\boldsymbol{x}_t, t, y)$ comprising four layers, each with 1024 units. Each layer employs the Swish activation function, $\text{Swish}(z) = z\sigma(z)$, where $\sigma(z)$ denotes the sigmoid function. The MLP is trained using the Adam optimizer for 100 epochs with a learning rate of 0.001. During each epoch, we sample $n_e = 8$ synthetic functions from the Gaussian process (the total number of synthetic functions is $n_g = n_e \times E = 8 \times 100 = 800$ functions) and generate $n_p = 1024$ samples for each function. At the testing phase, we sample high-value design candidates from the 128 best designs in the offline dataset, using $T = 200$ sequential denoising steps. All hyperparameters for modeling, training, and sampling with our model are summarized in Table 10.

Table 10: Hyperparameters for the generalized Brownian Bridge diffusion process in **ROOT** .

| | Hyperparameter | Value |
|---|---|---|
| Architecture | Hidden size | 1024 |
| | Number of layers | 4 |
| | Activation | Swish |
| Training | Number of epochs ($E$) | 100 |
| | Number of functions ($n_e$) (per epoch) | 8 |
| | Number of data points ($n_p$) (per function) | 1024 |
| | Learning rate | 0.001 |
| | Optimizer | Adam |
| | Batch size | 64 |
| | Conditional dropout ($\rho$) | 0.15 |
| Sampling | $\alpha$ | 0.8 |
| | $\beta$ | -1.5 |
| | Denoising steps | 200 |

## C   Additional Experiment Results

### C.1   Computation Resource

All our experiments were conducted on a system with the following specifications: Ubuntu 20.04.5, a single NVIDIA A100-SXM4-80GB GPU, and CUDA 10.1. Notably, our method requires less than 6GB of GPU memory per run.

### C.2   Additional Performance Evaluation of ROOT

In the main text, we have reported the 100th percentile scores. In this section, we present additional evaluation results at $80^{\text{th}}$ and $50^{\text{th}}$ percentiles, providing further insights into the performance distribution of our **ROOT**.

### C.2.1   Performance Evaluation at $80^{\text{th}}$ Percentile

As shown in Table 11, our method **ROOT** consistently demonstrates strong performance at the $80^{\text{th}}$ percentile level, achieving the best mean rank, i.e., **1.5**. Notably, in the **Ant** and **D' Kitty** tasks, **ROOT** achieves significant improvements over all baselines with remarkable score differences of **0.098** and **0.025** over the runner-ups in those tasks, respectively. **ROOT** also achieves the best result in

**TF-Bind-8**, outperforming the runner-up with a margin of **0.011**. In the **TF-Bind-10**, **ROOT** performs competitively to the best baselines. Overall, these results demonstrate the consistent reliability and stability of our model's performance.

Table 11: Experiments on Design-Bench Tasks. We report $80^{\text{th}}$ percentile score among $Q = 128$ candidates. Blue denotes the best entry in the column, and Brown denotes the second best. **Mean Rank** means the average rank of the method over all the experiment benchmarks.

| | Benchmarks | | | | Mean Rank |
|---|---|---|---|---|---|
| **Method** | **Ant** | **D'Kitty** | **TFBind8** | **TFBind10** | |
| $D_o$ (best) | 0.565 | 0.884 | 0.439 | 0.467 | - |
| BO-qEI | 0.629 ± 0.000 | 0.884 ± 0.000 | 0.439 ± 0.000 | 0.510 ± 0.011 | 11.00 / 15 |
| CMA-ES | 0.007 ± 0.013 | 0.718 ± 0.001 | 0.652 ± 0.017 | 0.543 ± 0.013 | 9.50 / 15 |
| REINFORCE | 0.182 ± 0.017 | 0.562 ± 0.197 | 0.622 ± 0.030 | 0.519 ± 0.007 | 11.25 / 15 |
| GA | 0.189 ± 0.014 | 0.762 ± 0.036 | 0.828 ± 0.027 | 0.516 ± 0.004 | 8.75 / 15 |
| COMs | 0.635 ± 0.031 | 0.887 ± 0.004 | 0.738 ± 0.027 | 0.526 ± 0.012 | 5.25 / 15 |
| CbAS | 0.542 ± 0.034 | 0.813 ± 0.012 | 0.585 ± 0.030 | 0.517 ± 0.008 | 10.25 / 15 |
| MINs | 0.746 ± 0.011 | 0.908 ± 0.004 | 0.545 ± 0.031 | 0.519 ± 0.010 | 6.50 / 15 |
| RoMA | 0.298 ± 0.033 | 0.738 ± 0.018 | 0.661 ± 0.010 | 0.525 ± 0.003 | 9.00 / 15 |
| DDOM | 0.749 ± 0.029 | 0.865 ± 0.009 | 0.526 ± 0.017 | 0.506 ± 0.004 | 9.75 / 15 |
| ICT | 0.708 ± 0.019 | 0.898 ± 0.004 | 0.667 ± 0.035 | 0.525 ± 0.016 | 6.00 / 15 |
| Tri-mentoring | 0.722 ± 0.015 | 0.902 ± 0.003 | 0.683 ± 0.047 | 0.531 ± 0.007 | 4.00 / 15 |
| GTG | 0.725 ± 0.077 | 0.913 ± 0.007 | 0.611 ± 0.038 | 0.506 ± 0.006 | 7.75 / 15 |
| LTR | 0.679 ± 0.016 | 0.902 ± 0.002 | 0.734 ± 0.025 | 0.508 ± 0.010 | 7.00 / 15 |
| GABO | 0.016 ± 0.009 | 0.705 ± 0.002 | 0.676 ± 0.021 | 0.512 ± 0.008 | 11.25 / 15 |
| **ROOT (ours)** | 0.847 ± 0.005 | 0.938 ± 0.002 | 0.839 ± 0.015 | 0.526 ± 0.007 | 1.50 / 15 |

### C.2.2 Performance Evaluation at $50^{\text{th}}$ Percentile

Table 12 reports the results achieved by all baselines at 50-th percentile. Consistent with our earlier observations, **ROOT** again achieves the best mean rank of **2.0**. In particular, we surpass the runner-up baseline, Gradient Ascent (GA), in the **TF-Bind-8** task with a significant score difference of **0.072**, securing the top rank among all other methods. In the **Ant** and **D'Kitty** tasks, **ROOT** also achieves substantial score gaps of **0.067** and **0.018** over the corresponding runner-ups, respectively. Furthermore, the reported standard deviations of **ROOT** in those tasks are **0.014** and **0.003** which are much lower than those of other baselines. This further ascertains the stability and superior performance of our method across the evaluation benchmark.

### C.3 Score Distribution of ROOT

To provide a more holistic comparison between the score distribution of **ROOT** and those of others, we collect the design candidates obtained from 8 runs ($128 \times 8 = 1024$ designs) to plot the distribution of scores achieved by **ROOT** and several other representative baselines. The results are shown in Fig. 2. Due to the wide range of scores produced by CMA-ES on the **Ant** task, we separately plot and compare its score distribution against **ROOT**. The remaining plots visualize and compare the score distributions of **ROOT** and 5 representative baselines. In each plot, we annotate the max and median lines to ease the visual comparison. In particular, Fig 2 reveals that **ROOT** often has a score distribution skewed towards higher-value regions, especially in the **D'Kitty** task. In the **Ant** task, although **ROOT** 's max score is not as high as that of CMA-ES, CMA-ES only produces a single good design while the rest (in its solution set) perform poorly, as also observed in Table 11 and Table 12. Otherwise, it can be observed consistently across all plots **ROOT** achieves better score distributions compared to other baselines.

**Summary.** In conclusion, all reported results have demonstrated consistently that our proposed method **ROOT** is, on average, the best-performing baseline in both score distributions and targeted evaluation percentiles. This consistent performance not only highlights the effectiveness and stability of **ROOT** but also reaffirms the robustness of our approach across a wide range of tasks.

Table 12: Experiments on Design-Bench Tasks. We report $50^{\text{th}}$ percentile score among $Q = 128$ candidates. Blue denotes the best entry in the column, and Brown denotes the second best. **Mean Rank** means the average rank of the method over all the experiment benchmarks.

| | | Benchmarks | | | | |
|---|---|---|---|---|---|---|
| **Method** | | **Ant** | **D'Kitty** | **TFBind8** | **TFBind10** | **Mean Rank** |
| $D_o$ (best) | | 0.565 | 0.884 | 0.439 | 0.467 | - |
| BO-qEI | | 0.569 ± 0.000 | 0.883 ± 0.000 | 0.439 ± 0.000 | 0.469 ± 0.005 | 7.75 / 15 |
| CMA-ES | | -0.043 ± 0.007 | 0.674 ± 0.016 | 0.536 ± 0.012 | 0.490 ± 0.015 | 8.75 / 15 |
| REINFORCE | | 0.140 ± 0.026 | 0.510 ± 0.203 | 0.450 ± 0.024 | 0.470 ± 0.010 | 11.00 / 15 |
| GA | | 0.137 ± 0.014 | 0.591 ± 0.132 | 0.603 ± 0.045 | 0.469 ± 0.006 | 8.75 / 15 |
| COMs | | 0.471 ± 0.034 | 0.862 ± 0.003 | 0.598 ± 0.031 | 0.475 ± 0.010 | 5.75 / 15 |
| CbAS | | 0.369 ± 0.008 | 0.748 ± 0.016 | 0.441 ± 0.021 | 0.465 ± 0.006 | 10.75 / 15 |
| MINs | | 0.618 ± 0.016 | 0.889 ± 0.003 | 0.421 ± 0.017 | 0.467 ± 0.010 | 7.25 / 15 |
| RoMA | | 0.224 ± 0.020 | 0.545 ± 0.170 | 0.519 ± 0.073 | 0.518 ± 0.003 | 8.50 / 15 |
| DDOM | | 0.568 ± 0.066 | 0.814 ± 0.016 | 0.404 ± 0.012 | 0.456 ± 0.002 | 11.00 / 15 |
| ICT | | 0.554 ± 0.018 | 0.872 ± 0.007 | 0.557 ± 0.031 | 0.457 ± 0.033 | 8.25 / 15 |
| Tri-mentoring | | 0.572 ± 0.016 | 0.884 ± 0.001 | 0.562 ± 0.051 | 0.475 ± 0.009 | 4.25 / 15 |
| GTG | | 0.645 ± 0.098 | 0.901 ± 0.005 | 0.460 ± 0.032 | 0.452 ± 0.010 | 7.25 / 15 |
| LTR | | 0.568 ± 0.016 | 0.885 ± 0.002 | 0.566 ± 0.026 | 0.466 ± 0.011 | 6.00 / 15 |
| GABO | | -0.039 ± 0.004 | 0.674 ± 0.005 | 0.496 ± 0.011 | 0.457 ± 0.008 | 11.50 / 15 |
| **ROOT (ours)** | | 0.712 ± 0.014 | 0.919 ± 0.003 | 0.675 ± 0.026 | 0.473 ± 0.004 | 2.00 / 15 |

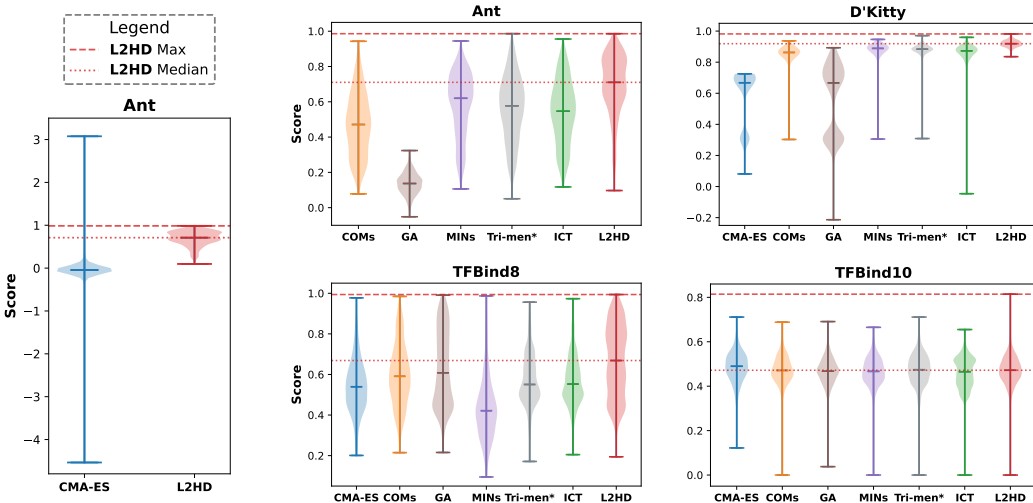

Figure 2: Score distribution of found candidates of **ROOT** compared to others. To avoid cluttering the plots with long chunks of text, we use the abbreviation **Tri-men\*** to denote **Tri-mentoring**.

## C.4 Computational Complexity Analysis

**Computational Complexity.** Computational cost of **ROOT** is as follows:

- For each training epoch: Fitting $n_f$ Gaussian processes requires $O(n_f n^3)$ where $n$ is the offline data size. Performing M gradient steps (querying the GP's mean function) to generate $n_p$ synthetic points from each of $n_f$ functions requires $O(n_f M n_p n)$. Training the BBDM model requires T diffusion steps per sample, where each step involves a forward and backward pass through the score network $\epsilon_\theta$, giving a total cost of $O(T(f_\theta + b_\theta)n_f n_p)$, where $f_\theta, b_\theta$ denote the cost of forward and backward, respectively.

- For the whole training process, the overall cost is $O(E(n_f n^3 + n_f M n_p n + T(f_\theta + b_\theta)n_f n_p))$ where $E$ is the number of epochs.

- For the inference process, the overall cost is $O(QDf_\theta)$ where $Q$ and $D$ are the number of selected candidates and denoising steps.

Therefore, the total computational cost of **ROOT** is:

$$O(E(n_f n^3 + n_f M n_p n + T(f_\theta + b_\theta) n_f n_p)) + O(QDf_\theta)$$

Furthermore, we note that the most complex term in ROOT's computational cost, GP fitting, can be further mitigated by using sparse GP techniques [49], which scales linearly in the number of inputs.

**Empirical Runtime.** We empirically evaluate the runtime of our method and compare it against several baselines across a range of tasks. As shown in Table 13, the results demonstrate that **ROOT** achieves faster execution time than several widely used baselines, including CMA-ES, MINS, and Tri-mentoring, while maintaining competitive or superior performance. All experiments are conducted on a **single GPU**, with each task requiring approximately **4GB** of memory.

Table 13: Runtime (in seconds) of different models across four tasks.

| Model | Ant | D'Kitty | TF-Bind-8 | TF-Bind-10 |
|---|---|---|---|---|
| GA | 122 | 200 | 124 | 420 |
| BO-QEI | 219 | 310 | 402 | 383 |
| CMA-ES | 5747 | 4889 | 2667 | 4323 |
| MINS | 797 | 998 | 2213 | 792 |
| REINFORCE | 240 | 257 | 507 | 373 |
| CBAS | 394 | 390 | 835 | 529 |
| ICT | 188 | 189 | 216 | 639 |
| Tri-mentoring | 4276 | 3921 | 4673 | 3410 |
| DEMO | 668 | 1489 | 1024 | 1696 |
| **ROOT** | 298 | 297 | 407 | 575 |

## C.5 Effectiveness of Gaussian process for Generating Synthetic Data.

### C.5.1 DNN-based Approach for Generating Synthetic Data

There are several strategies for sampling closed-form functions fitted to offline data (to provide functions that are similar to the oracle). In **ROOT**, we adopt a standard Gaussian process (GP) approach due to its efficiency, ease of implementation, and strong empirical performance. GPs are particularly advantageous for their simplicity in debugging and their effectiveness in modeling predictive uncertainty. Our motivation is further motivated by ExPT [45], a recent few-shot offline optimization baseline, which similarly employs GPs to generate pseudo labels for offline data.

While alternative generative models exist, they often introduce additional complexity and computational overhead. For instance, we conducted experiments using deep neural networks (DNNs) in place of GPs. One such approach involves randomly initializing the weights of a DNN and training it to fit the offline dataset, then serving as a single sampled function. However, this approach is computationally expensive. Generating 800 such functions using DNNs, each requiring 800 separate training runs, takes approximately **100** minutes, compared to under **300** seconds for the entire training process using our GP-based method.

To further investigate this, we evaluated two less costly alternative setups: (1) an ensemble of five DNNs replacing all 800 GP functions, denoted as **DNN5**, and (2) a single DNN replacing all 800 GP sampled functions, denoted as **DNN1**. As shown in the Table 14, both alternatives perform worse than our GP-based approach and incur significantly higher computational costs.

### C.5.2 Bins-based Approach for Generating Synthetic Data

Beyond the DNN-based approach, we also explore two other heuristic methods that do not depend on Gaussian processes (GP). The first method, *2 bins*, partitions the offline data into two bins, the lowest 50th percentile and the highest 50th percentile, and samples low- and high-value designs $(X^-, X^+)$ from the corresponding bins respectively. The second method, *64 bins*, divides the data into 64

Table 14: Performance and runtime of different methods for sampling function on the **Ant** task.

| Model | Performance | Time (s) |
|-------|-------------|----------|
| DNN5 | 0.637 ± 0.268 | 1297 |
| DNN1 | 0.630 ± 0.201 | 751 |
| **ROOT** | 0.965 ± 0.014 | 298 |

bins based on the $y$ values and then samples $X^-$ from the lowest bin and $X^+$ from the highest. This approach is similar to the sampling strategy in [38], where trajectories with increasing outputs are selected from offline data to train a model that progressively guides designs from the lowest to the highest bin. In addition, we also examine another GP-based approach that utilizes only one GP function to generate synthetic data. As reported in Table 15, the GP-based methods consistently outperform the above bin-based heuristic approaches, with GP (800 functions) achieving the best performance.

Table 15: Effectiveness of Gaussian process for generating synthetic data.

| | Type | Ant | D'Kitty | TF-Bind-8 | TF-Bind-10 |
|-------|------|-----|---------|-----------|------------|
| No-GP | 2 bins | 0.745 ± 0.097 | 0.952 ± 0.007 | 0.775 ± 0.057 | 0.641 ± 0.034 |
| | 64 bins | 0.941 ± 0.020 | 0.952 ± 0.009 | 0.837 ± 0.055 | 0.651 ± 0.054 |
| GP | GP (1 function) | 0.955 ± 0.013 | 0.971 ± 0.004 | 0.984 ± 0.012 | 0.657 ± 0.029 |
| | GP (800 functions) | 0.965 ± 0.014 | 0.972 ± 0.005 | 0.986 ± 0.007 | 0.685 ± 0.053 |

## C.6 GP Hyperparameters Sampling Methods

To diversify the GP mean functions, various approaches can be used to sample GP hyperparameters. In our work, we adopt a simple strategy by uniformly sampling the GP hyperparameters, i.e., the lengthscale $\ell_s$ and variance $\sigma_s$, from a fixed range, as presented in line 5 of Algorithm 1. This approach follows the prior practice in ExPT [45], which also samples these hyperparameters uniformly from a small range and achieves strong performance.

To explore alternative strategies, we have conducted additional experiments. In a **MAP-based** setting, we first optimize the hyperparameters $\ell_0$ and $\sigma_0$ by minimizing the negative log marginal likelihood (NLML) on $D_o = \{\boldsymbol{x}_i, y_i\}_{i=1}^n$, which is commonly used in Gaussian processes:

$$\mathcal{L}_{\text{NLML}}(\phi) \;=\; \frac{1}{2}\mathbf{y}^\top K_\phi^{-1}\mathbf{y} \;+\; \frac{1}{2}\log|K_\phi| \;+\; \frac{n}{2}\log 2\pi \;,$$

where $K_\phi$ is the kernel matrix parameterized by $\phi = \{\ell_0, \sigma_0\}$, and $n$ is the number of training data points. After obtaining these MAP estimates, we sample new hyperparameters uniformly around them: $\ell_s \sim [\ell_0 - \delta, \ell_0 + \delta]$ and $\sigma_s \sim [\sigma_0 - \delta, \sigma_0 + \delta]$, instead of setting $\ell_0, \sigma_0$ as in Table 9.

In addition, we have also experimented with an **MCMC-based** approach to sample these hyperparameters. In this setting, we first define a prior distribution $p(\ell_s, \sigma_s)$ (e.g., a Gaussian prior) and a likelihood $p(D_o \mid \ell_s, \sigma_s)$, derived similarly from the marginal likelihood. We then apply MCMC sampling to draw $n_g$ hyperparameter combinations from the posterior distribution, i.e $p(\ell_s, \sigma_s \mid D_o) \propto p(D_o \mid \ell_s, \sigma_s) \times p(\ell_s, \sigma_s)$.

As shown in Table 16, our uniform sampling method consistently outperforms the aforementioned alternatives. Also, both alternatives introduce significant computational overhead. In particular, the MCMC-based approach requires substantial runtime, incurring approximately **1000** seconds per run, even when using only 8 warmup steps. We leave further exploration of more sophisticated hyperparameter sampling strategies to future work.

## C.7 Hyper-parameter Tuning

**Number of GP mean functions**. We additionally conducted experiments on the number of GP mean functions to generate synthetic data. In our original method, we use $n_g = n_e \times M = 800$ functions; here, we vary this number from 100 to 1000. The empirical results in Table 17 indicate that the final

Table 16: Performance and runtime of different GP hyperparameter sampling methods on **Ant** task.

| Method | Performance |
|---|---|
| MAP ($n_g = 800$) | $0.940 \pm 0.023$ |
| MCMC ($n_g = 200$) | $0.884 \pm 0.017$ |
| MCMC ($n_g = 400$) | $0.921 \pm 0.010$ |
| MCMC ($n_g = 600$) | $0.932 \pm 0.016$ |
| MCMC ($n_g = 800$) | $0.916 \pm 0.009$ |
| **ROOT** ($n_g = 800$) | $0.965 \pm 0.014$ |

performance improves as the number of Gaussian Processes (GPs) increases. However, there may exist a saturation point (e.g., 800 GPs), beyond which further increasing the number of GPs could lead to a decline in performance.

Table 17: Performance on **Ant** and **TF-Bind-10** with varying number of GPs.

| #GPs | Ant | TF-Bind-10 |
|---|---|---|
| 100 | $0.914 \pm 0.015$ | $0.632 \pm 0.020$ |
| 200 | $0.914 \pm 0.015$ | $0.677 \pm 0.038$ |
| 400 | $0.959 \pm 0.010$ | $0.631 \pm 0.018$ |
| 600 | $0.956 \pm 0.021$ | $0.674 \pm 0.044$ |
| **800 (ROOT)** | $0.965 \pm 0.014$ | $0.685 \pm 0.053$ |
| 1000 | $0.960 \pm 0.016$ | $0.662 \pm 0.033$ |

**GP hyperparameters**. We also employed an ablation study to see the effect of Gaussian hyperparameters (i.e., initial lengthscale $\ell_0$) on our final results. As can be seen clearly from the Fig. 3, the optimal range for $\ell_0$ yields around $1.0$ for the continuous tasks and $6.25$ for the discrete tasks.

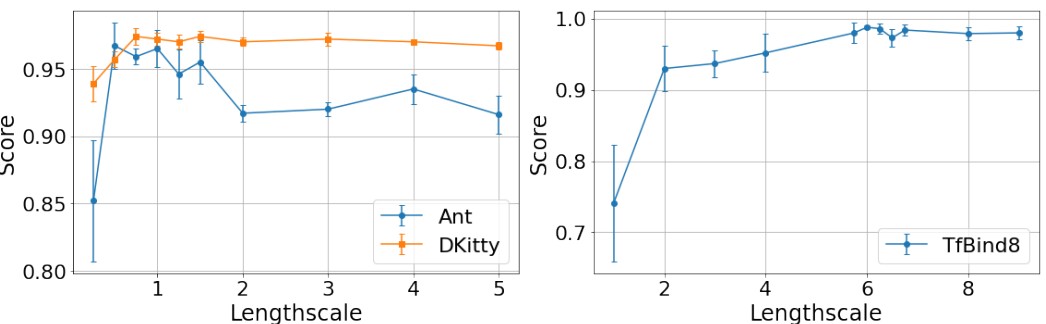

Figure 3: Effect of lengthscale on performance for Ant and DKitty.

**GP hyperparameters range size $\delta$ and step size $\eta$**. In the data-generation phase, we uniformly sample the Gaussian process lengthscale and variance from the intervals $[\ell_0 - \delta, \; \ell_0 + \delta]$ and $[\sigma_0 - \delta, \; \sigma_0 + \delta]$, respectively (see Line 5 Algorithm 1). Choosing $\delta$ is a trade-off: if it is too large, the probabilistic bridge will learn on functions that deviate excessively from the oracle, but if it is too small, we lose the diversity needed for robust generalization. We conducted a sweep over different values of $\delta$; as shown in Table 18, $\delta = 0.25$ achieves the best performance across both the **Ant** and **TF-Bind-8** tasks. In addition, during this phase we apply $M$ gradient steps with fixed step size $\eta$ (see Eq.12 and Eq. 13). Our experiments in Table 19 demonstrate that for continuous tasks, setting $\eta = 0.001$ achieves the best score.

**GP kernel choice**. We experimented with the Matern kernel for the GP-based synthetic data generation, as reported in Table 20. The results indicate that the commonly used RBF kernel consistently yields better performance, supporting our decision to use it as the default.

**Incorporating explicit uncertainty via acquisition functions**. In Gaussian Process literature, incorporating explicit uncertainty via acquisition functions like UCB or LCB is a promising approach

Table 18: Performance on **Ant** and **TF-Bind-8** with varying $\delta$.

| $\delta$ | Ant | TF-Bind-8 |
|---|---|---|
| 0.05 | $0.953 \pm 0.011$ | $0.982 \pm 0.005$ |
| 0.10 | $0.957 \pm 0.014$ | $0.986 \pm 0.006$ |
| **0.25** (ours) | $0.965 \pm 0.014$ | $0.986 \pm 0.007$ |
| 0.50 | $0.959 \pm 0.013$ | $0.981 \pm 0.007$ |
| 1.00 | $0.959 \pm 0.014$ | $0.989 \pm 0.003$ |

Table 19: Performance on **Ant** and **D'Kitty** with varying $\eta$.

| $\eta$ | Ant | D'Kitty |
|---|---|---|
| 0.0005 | $0.969 \pm 0.017$ | $0.971 \pm 0.003$ |
| 0.00075 | $0.968 \pm 0.013$ | $0.969 \pm 0.005$ |
| **0.001** (ours) | $0.965 \pm 0.014$ | $0.972 \pm 0.005$ |
| 0.0025 | $0.964 \pm 0.010$ | $0.976 \pm 0.004$ |
| 0.005 | $0.948 \pm 0.011$ | $0.977 \pm 0.003$ |

Table 20: Performance comparison of different GP kernels on **Ant** and **TF-Bind-8**.

| Kernel | Ant | TFBind8 |
|---|---|---|
| Matern | $0.966 \pm 0.013$ | $0.848 \pm 0.055$ |
| RBF | $0.965 \pm 0.014$ | $0.986 \pm 0.007$ |

Table 21: Performance of **ROOT** when replacing the GP's mean function with UCB and LCB.

| Method | Ant | TF-Bind-8 |
|---|---|---|
| UCB | $0.964 \pm 0.017$ | $0.966 \pm 0.017$ |
| LCB | $0.949 \pm 0.008$ | $0.961 \pm 0.025$ |
| **ROOT** | $0.965 \pm 0.014$ | $0.986 \pm 0.007$ |

for balancing exploration and exploitation. To evaluate this, we conducted a small-scale experiment replacing the GP's mean function with UCB and LCB scores to generate synthetic data. The results, shown in Table 21, indicate that while UCB and LCB perform reasonably well, they do not outperform our original approach based on the GP's mean function. We believe this is because uncertainty has already been implicitly captured through a different mechanism, by sampling multiple mean functions from a population of posterior GPs trained on the same offline dataset using different kernel configurations.

## C.8 Score and Pseudo-Value Distribution of Generated Samples from GP

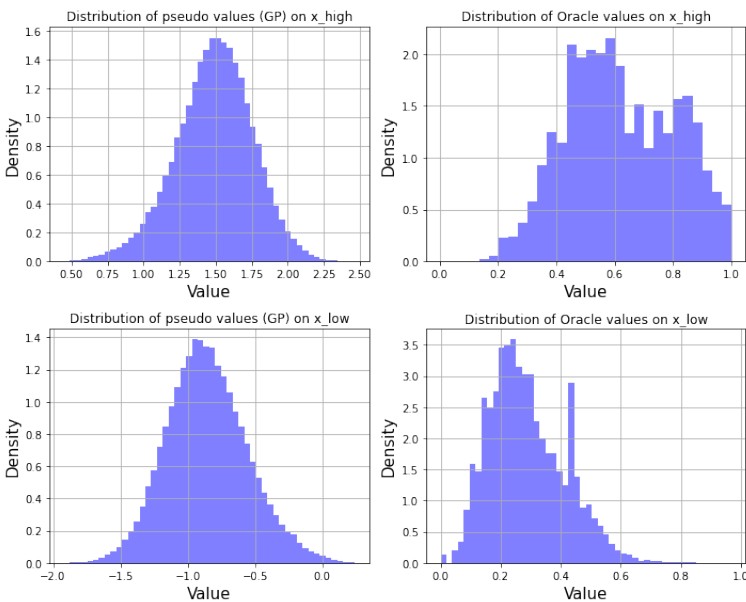

Figure 4: Distribution of pseudo-values (GP) and oracle values on low- and high-value regions.

To further demonstrate the efficiency of Gaussian processes, we conducted an experiment on the **Ant** task using low and high regions produced by 800 GP mean functions. Each function produces 1024 low-value points ($X^-$ or x_low) along with 1024 high-value points ($X^+$ or x_high). We concatenate all $1024 \times 800$ points to form the distribution for the low-value and high-value designs. We then

plotted both the oracle value distribution by using the oracle function to evaluate and the pseudo-value distribution ($y^-$ and $y^+$ predicted by GP). As shown in Figure 4, the low and high regions correspond to two clearly separated distributions. The oracle values in $\boldsymbol{X}^+$ are skewed toward higher values, while those in $\boldsymbol{X}^-$ are skewed toward lower values. This separation facilitates effective training of our probabilistic bridge model as it learns to transform between two distinct distributions.

### C.9 Limited Budget Settings

While a budget of $Q = 128$ candidates is a commonly adopted setting, we also conduct experiments under more constrained budgets to evaluate the robustness of our method. The results, shown in Table 22, demonstrate that **ROOT** continues to achieve strong performance even in these highly limited-query scenarios, outperforming existing baselines.

Table 22: Performance of different methods under extremely limited offline data budgets ($Q$). **ROOT** consistently outperforms baselines on both **Ant** and **TF-Bind-8**.

| Q Budget | Method | Ant | TFBind8 |
|---|---|---|---|
| 16 | GA | 0.250 ± 0.033 | 0.938 ± 0.020 |
| | COMs | 0.789 ± 0.070 | 0.917 ± 0.059 |
| | **ROOT** | 0.938 ± 0.022 | 0.956 ± 0.028 |
| 32 | GA | 0.252 ± 0.036 | 0.961 ± 0.024 |
| | COMs | 0.776 ± 0.033 | 0.871 ± 0.030 |
| | **ROOT** | 0.942 ± 0.019 | 0.974 ± 0.020 |
| 64 | GA | 0.281 ± 0.023 | 0.978 ± 0.020 |
| | COMs | 0.834 ± 0.051 | 0.922 ± 0.036 |
| | **ROOT** | 0.942 ± 0.020 | 0.974 ± 0.021 |

### C.10 Strong measurement noise setting

Although ROOT uses synthetic labels from deterministic GP means, it remains robust to strong measurement noise. This is due to the use of a diverse GP ensemble with varied hyperparameters, which implicitly captures a wide range of uncertainties present in the offline dataset. Sampling from this ensemble produces synthetic data that reflects variability in the objective, helping the bridge model generalize more effectively. To validate this, we ran additional experiments on **TF-Bind-8** with label perturbed by Gaussian noise $\mathbb{N}(0, \epsilon)$. As shown in the Table 23, **ROOT** maintains strong performance and outperforms other baselines across noise levels, demonstrating its resilience to measurement noise.

Table 23: Performance of different methods on **TF-Bind-8** task with label perturbed by measurement noise $\mathbb{N}(0, \epsilon)$.

| $\epsilon$ | 0.01 | 0.1 | 0.2 |
|---|---|---|---|
| GA | 0.967 ± 0.015 | 0.970 ± 0.006 | 0.963 ± 0.010 |
| COMs | 0.956 ± 0.024 | 0.925 ± 0.034 | 0.952 ± 0.013 |
| REINFORCE | 0.947 ± 0.030 | 0.933 ± 0.025 | 0.930 ± 0.042 |
| **ROOT** | 0.969 ± 0.016 | 0.971 ± 0.007 | 0.965 ± 0.014 |

### C.11 High-dimensional continuous task

In the Design-Bench benchmark, the **Hopper** task represents a high-dimensional continuous task with 5,126 input dimensions. Although prior work has noted a highly noisy and inaccuracy oracle function for this task, we conducted a small experiment to evaluate **ROOT**'s performance on **Hopper** as a test of scalability. As shown in the Table 24, **ROOT** outperforms some representative baselines, demonstrating its ability to effectively handle high-dimensional design spaces.

Table 24: Performance of different methods on the Hopper Controller task.

| Method | Hopper Controller |
|---|---|
| GA | -0.068 ± 0.001 |
| MINs | 0.267 ± 0.350 |
| Reinforce | -0.009 ± 0.067 |
| **ROOT** | 0.541 ± 0.042 |

Table 25: Performance of **ROOT** with different bridge configuration across the benchmark tasks.

| Bridges | Ant | D'Kitty | TF-Bind-8 | TF-Bind-10 |
|---|---|---|---|---|
| **ROOT** (BBDM) | 0.965 ± 0.014 | 0.972 ± 0.005 | 0.986 ± 0.007 | 0.685 ± 0.053 |
| **ROOT** (OUDM) | 1.940 ± 0.599 | 0.723 ± 0.001 | 0.927 ± 0.049 | 0.675 ± 0.124 |

## D   Other Practical Setups for Our Probabilistic Bridge Framework

In this paper, we employ the Brownian bridge as our practical probabilistic bridge (see Appendix B.2), which achieves significant performance across our experiments. However, our proposal method is a flexible framework that can be universally adapted to other probabilistic bridges. In this section, we will demonstrate another design: **Ornstein-Uhlenbeck Bridge**.

From our definition of the probabilistic bridge in Section 3.1.1, we can choose $\psi_t$ and $\kappa_{t,k}$ as

$$\psi_t(\boldsymbol{x}_0, \boldsymbol{x}_T) \;=\; \boldsymbol{x}_0 \cdot \frac{\sinh(\alpha(T-t))}{\sinh(\alpha T)} + \boldsymbol{x}_T \cdot \frac{\sinh(\alpha t)}{\sinh(\alpha T)} \;, \tag{26}$$

$$\kappa_{t,k} \;=\; \frac{\sinh(\alpha \cdot \min(t,k)) \cdot \sinh(\alpha \cdot (T - \max(t,k)))}{\alpha \cdot \sinh(\alpha T)} \;. \tag{27}$$

This choice will result in the formula of the Ornstein-Uhlenbeck bridge [1] with the hyperparameter $\alpha$. Once this bridge is learned, we can simulate $\boldsymbol{x}_t$ following the previously derived simulation approach in Eq. 5 which, under the OU instantiation, becomes,

$$\boldsymbol{x}_t \;=\; \boldsymbol{x}_0 \cdot \frac{\sinh(\alpha(T-t))}{\sinh(\alpha T)} \;+\; \boldsymbol{x}_T \cdot \frac{\sinh(\alpha t)}{\sinh(\alpha T)} \;+\; \sqrt{\frac{\sinh(\alpha t) \cdot \sinh(\alpha(T-t))}{\alpha \cdot \sinh(\alpha T)}} \cdot \boldsymbol{\epsilon}_t \tag{28}$$

where $\boldsymbol{\epsilon}_t \sim \mathbb{N}(0, \boldsymbol{I})$. From the above equation, we can also represent $\boldsymbol{x}_0$ in terms of $\boldsymbol{x}_t$ and $\boldsymbol{x}_T$:

$$\boldsymbol{x}_0 \;=\; \left( \boldsymbol{x}_t - \boldsymbol{x}_T \cdot \frac{\sinh(\alpha t)}{\sinh(\alpha T)} - \sqrt{\frac{\sinh(\alpha t) \cdot \sinh(\alpha(T-t))}{\alpha \cdot \sinh(\alpha T)}} \cdot \boldsymbol{\epsilon}_t \right) \cdot \frac{\sinh(\alpha T)}{\sinh(\alpha(T-t))} \;. \tag{29}$$

The transition probability of the localized bridge/flow, i.e., $q(\boldsymbol{x}_{t-1} \mid \boldsymbol{x}_t, \boldsymbol{x}_0, \boldsymbol{x}_T)$ in Eq. 6, can be derived by using the conditional Gaussian rule which results in the following transition mean,

$$\boldsymbol{\mu}(\boldsymbol{x}_t, \boldsymbol{x}_0, \boldsymbol{x}_T) \;=\; \psi_{t-1}(\boldsymbol{x}_0, \boldsymbol{x}_T) \;+\; \kappa_{t-1,t} \kappa_{t,t}^{-1} (\boldsymbol{x}_t - \psi_t(\boldsymbol{x}_0, \boldsymbol{x}_T)) \;. \tag{30}$$

Substituing $\boldsymbol{x}_t = \psi_t(\boldsymbol{x}_0, \boldsymbol{x}_T) + \sqrt{\kappa_{t,t}} \cdot \boldsymbol{\epsilon}_t$, the above transition mean can be rewritten as:

$$
\begin{aligned}
\boldsymbol{\mu}(\boldsymbol{x}_t, \boldsymbol{x}_0, \boldsymbol{x}_T) \;=&\; \psi_{t-1}(\boldsymbol{x}_0, \boldsymbol{x}_T) + \kappa_{t-1,t} \kappa_{t,t}^{-1/2} \boldsymbol{\epsilon}_t \\
=&\; \boldsymbol{x}_0 \frac{\sinh\big(\alpha\,(T-t+1)\big)}{\sinh(\alpha T)} + \boldsymbol{x}_T \frac{\sinh\big(\alpha\,(t-1)\big)}{\sinh(\alpha T)} \\
&+\; \boldsymbol{\epsilon}_t \frac{\sinh\big(\alpha\,(t-1)\big) \sinh\big(\alpha\,(T-t)\big)}{\sqrt{\alpha\,\sinh(\alpha T)\,\sinh(\alpha t)\,\sinh(\alpha(T-t))}} \;.
\end{aligned}
$$
(31)

(32)

By substituting $\boldsymbol{x}_0$ from Eq. 29, we get a closed-form formula for $\boldsymbol{\mu}(\boldsymbol{x}_t, \boldsymbol{x}_0, \boldsymbol{x}_T) = u_t \cdot \boldsymbol{x}_t + v_t \cdot \boldsymbol{x}_T + w_t \cdot \boldsymbol{\epsilon_t}$ where the coefficients are computed as

$$u_t = \frac{\sinh\big(\alpha\,(T-t+1)\big)}{\sinh\big(\alpha\,(T-t)\big)}, \quad v_t = \left[\frac{\sinh\big(\alpha\,(t-1)\big)}{\sinh(\alpha T)} - \frac{\sinh\big(\alpha\,(T-t+1)\big)\,\sinh(\alpha t)}{\sinh(\alpha T)\,\sinh\big(\alpha\,(T-t)\big)}\right], \tag{33}$$

$$w_t = \sqrt{\frac{\sinh\big(\alpha\,(T-t)\big)}{\alpha\,\sinh(\alpha T)\,\sinh(\alpha t)}} \cdot \left[\sinh\big(\alpha\,(t-1)\big) - \sinh(\alpha t) \cdot \frac{\sinh\big(\alpha\,(T-t+1)\big)}{\sinh\big(\alpha\,(T-t)\big)}\right]. \tag{34}$$

Next, following prior practice, we again reparameterize the target-agnostic transformation $p_\theta(\boldsymbol{x}_{t-1}|\boldsymbol{x}_t, \boldsymbol{x}_T)$ as a neuralized Gaussian with mean $\boldsymbol{\mu}_\theta(\boldsymbol{x}_t, \boldsymbol{x}_0, \boldsymbol{x}_T) = u_t \cdot \boldsymbol{x}_t + v_t \cdot \boldsymbol{x}_T + w_t + \epsilon_\theta(\boldsymbol{x}_t, \boldsymbol{x}_T, t)$ and same covariance matrix as the local transition, the (training) ELBO loss in Eq. 8 can be simplified as:

$$\boldsymbol{\theta}_{\mathrm{PB}} = \arg\min_{\boldsymbol{\theta}} \mathbb{E}_{(\boldsymbol{x}_0, \boldsymbol{x}_T) \sim D_s, \boldsymbol{\epsilon}_t \sim \mathbb{N}(0, \boldsymbol{I})} \Big\| \epsilon_\theta(\psi_t(\boldsymbol{x}_0, \boldsymbol{x}_T) + \kappa_{t,t} \cdot \epsilon_t, \boldsymbol{x}_T, t) - \epsilon_t \Big\|^2. \tag{35}$$

In addition, incorporating the classifier-free guidance techniques leads to:

$$\boldsymbol{\theta}_{\mathrm{PB}} = \arg\min_{\boldsymbol{\theta}} \mathbb{E}_{(\boldsymbol{x}_0, y_0, \boldsymbol{x}_T, y_T), \boldsymbol{\epsilon}_t} \Big\| \epsilon_\theta(\psi_t(\boldsymbol{x}_0, \boldsymbol{x}_T) + \kappa_{t,t} \cdot \epsilon_t, t, \gamma \cdot y + (1-\gamma) \cdot \emptyset) - \epsilon_t \Big\|^2, \tag{36}$$

where $\gamma \sim \mathrm{Ber}(\rho)$. Once the training process is done, we can employ the trained network $\epsilon_\theta$ for our optimization-via-simulation process:

$$\boldsymbol{x}_{t-1} = u_t \cdot \boldsymbol{x}_t + v_t \cdot \boldsymbol{x}_T + w_t \cdot \boldsymbol{\epsilon}_\theta(x_t, t, y) + \sqrt{\tilde{\kappa}_{t-1}} \cdot \boldsymbol{\epsilon}, \tag{37}$$

where $\tilde{\kappa}_{t-1} = \kappa_{t-1,t-1} - \kappa_{t-1,t}\kappa_{t,t}^{-1}\kappa_{t,t-1}$ and $\boldsymbol{\epsilon} \sim \mathbb{N}(0, \boldsymbol{I})$.

The performance of the OU-based **ROOT** is compared against the original Brownian-based **ROOT** in Table 25 . The results show that the OU-based variant of **ROOT** performs competitively to the original Brownian variant on **TF-Bind-8** and **TF-Bind-10**. On the **Ant** task, the OU-based variant is better while on the **D'Kitty** task, the Brownian variant is better. We leave a more thorough investigation across different variants of **ROOT** with different bridge configuration to future work as such detailed investigation is beyond the scope of our current work.

## E  Broader Impact and Limitation

**Broader Impact.**  This work provides a novel lens for offline black-box optimization by reframing it as a distributional translation problem, potentially inspiring new probabilistic modeling techniques in low-data regimes. The approach opens up new possibilities for data-efficient optimization in applications where function evaluations are expensive or infeasible. These include (but are not limited to) materials discovery, policy design, and automated experimentation. However, caution must be exercised when deploying such methods in safety-critical domains, as reliance on synthetic priors may introduce epistemic uncertainty that requires careful quantification and calibration.

**Limitation.**  One practical consideration is the additional computational overhead introduced by fitting multiple Gaussian processes to construct the synthetic function ensemble. While this step is performed offline and enables better data efficiency during optimization, it may require tuning and parallelization to scale effectively with large input dimensions or limited compute resources.

