# OpenReview forum: "ROOT: Rethinking Offline Optimization as Distributional Translation via Probabilistic Bridge"
_NeurIPS.cc/2025/Conference — NeurIPS 2025 spotlight_

### Official Review · Reviewer_tDV4 · 2025-06-04

**Clarity:** 4
**Significance:** 3
**Originality:** 3
**Rating:** 5
**Confidence:** 4

**Summary:**

This paper tackles offline black-box optimization by proposing a novel framework that treats the task as a distributional translation problem—from low-performing to high-performing inputs. Instead of relying solely on scarce offline data, it learns a probabilistic bridge using synthetic functions, constructed as averaged Gaussian processes fit to the data. This approach improves generalization and achieves state-of-the-art performance on benchmark tasks.

**Questions:**

See Strengths And Weaknesses

**Ethical Concerns:**

["NO or VERY MINOR ethics concerns only"]

**Final Justification:**

i recommend accept as the author addresses majority of my concerns.

**Limitations:**

See Strengths And Weaknesses

**Quality:**

3

**Strengths And Weaknesses:**

Strengths: The idea of leveraging synthetic data to learn a mapping from low- to high-performing inputs is compelling. The results are strong, and I particularly appreciate the focus on addressing the core challenge of offline black-box optimization—data scarcity. This paper offers a promising direction to mitigate that limitation.

Weaknesses: One potential concern is that the synthetic functions may not closely reflect real-world tasks, such as biological sequence design. Additionally, the paper appears to overlook a recent and relevant survey in this area: Offline Model-Based Optimization: Comprehensive Review (https://arxiv.org/abs/2503.17286).

---

> ### Author Rebuttal · Authors · 2025-07-31
>
> We truly appreciate the reviewer's thoughtful feedback and acceptance rating. We would like to address the reviewer's questions below:
>
> >One potential concern is that the synthetic functions may not closely reflect real-world tasks, such as biological sequence design
>
> We thank the reviewer for this thoughtful observation. We agree that Gaussian process (GP) priors may not fully capture the complexity of biological fitness landscapes, which can exhibit non-smooth behaviors that GPs are not well suited to model. That said, our use of GP-based synthetic functions is not intended to simulate the true oracle function in its entirety, but rather to approximate its local behavior within the region surrounding the offline data. By sampling a diverse set of posterior mean functions from GPs conditioned on real observations, we aim to reflect a broad range of plausible oracle behaviors near the data manifold. While this does not guarantee global fidelity to the true oracle, it provides a data-driven and computationally efficient means of reliably identifying improved designs within the local region supported by the offline data.
>
> To further validate the practical applicability of ROOT in real-world scenarios, we evaluated it on a biological sequence design task (RNA-binding), as shown in Table 2. The strong performance in this setting suggests that ROOT remains effective even when the true oracle is not fully consistent with a GP prior, highlighting its robustness in real-world applications.
>
> >Additionally, the paper appears to overlook a recent and relevant survey in this area: Offline Model-Based Optimization: Comprehensive Review (https://arxiv.org/abs/2503.17286).
>
> Thank you for bringing this valuable survey to our attention. We will revise the paper to include a discussion and citation of this in the related work section to ensure comprehensive coverage of key developments in the offline optimization literature.
>
> We hope that our responses have addressed your remaining questions satisfactorily. If you have any questions or points that require additional clarification, please feel free to let us know. We would be happy to continue the discussion and provide any additional details.

---

### Official Review · Reviewer_fMmR · 2025-07-02

**Clarity:** 3
**Significance:** 3
**Originality:** 3
**Rating:** 5
**Confidence:** 5

**Summary:**

This paper proposes ROOT, a new framework for offline black-box optimization that formulates the problem as a probabilistic bridge between a support distribution (from offline data) and an unknown optimal distribution. The authors frame offline optimization as learning a time-indexed trajectory in distribution space, connecting the initial offline data distribution to the target high-value design distribution. ROOT leverages variational inference to approximate this bridge and instantiates a diffusion-like architecture to model the intermediate transitions. The authors evaluate ROOT on offline molecule and biological sequence optimization tasks and compare against existing optimization baselines.

**Questions:**

- How robust is ROOT to poor offline data coverage? Have you tested performance under varying dataset support quality?

- Can you clarify the computational cost of ROOT compared to other diffusion-based energy-based approaches?

- Are there formal convergence or generalization guarantees that can support the ROOT framework theoretically?

**Ethical Concerns:**

["NO or VERY MINOR ethics concerns only"]

**Final Justification:**

The authors effectively addressed most of my concerns during the rebuttal phase, so I raised my score.

**Limitations:**

Yes

**Quality:**

3

**Strengths And Weaknesses:**

Strengths
- The idea of framing offline optimization as a probabilistic distributional bridge is creative and provides a new generative modeling perspective on offline design problems.

- The ROOT architecture is flexible and can incorporate different score functions (e.g., predictive surrogates) to guide optimization, making it potentially extensible across various problem domains.

- ROOT performs competitively on molecule and sequence tasks, demonstrating that the proposed formulation is viable in practice.

Weaknesses
- Theory–practice gap: The proposed variational formulation and bridge process are theoretically interesting but lack formal results (e.g., guarantees, convergence bounds).

- Architectural details underexplained: Some implementation choices (e.g., score function choices, transition structure) would benefit from more ablation or motivation.

---

> ### Author Rebuttal · Authors · 2025-07-31
>
> We really appreciate the reviewer’s thorough feedback with a positive rating, and would like to address the remaining concerns as follows:
> >Theory–practice gap: The proposed variational formulation and bridge process are theoretically interesting but lack formal results (e.g., guarantees, convergence bounds).
>
> and
>
> > Are there formal convergence or generalization guarantees that can support the ROOT framework theoretically?
>
> We appreciate the reviewer’s concern regarding the absence of formal guarantees or convergence bounds. This is indeed a limitation shared by standard diffusion-based approaches and remains an open challenge in the community. While we agree that developing theoretical foundations is important, this is a complex and ongoing area of research. A promising direction for future work is to relate the optimization trajectory of the oracle function to the underlying dynamics of the denoising diffusion process, though doing so will require substantial additional investigation.
>
> We note that most state-of-the-art offline optimization methods similarly lack formal guarantees, reflecting the broader difficulty of establishing theoretical bounds in this setting. Nonetheless, our method delivers strong empirical performance across multiple benchmarks and consistently compares favorably with existing approaches. We believe this demonstrates the practical utility of our framework, even in the absence of formal guarantees at this stage. We also note that our empirical evaluation includes a diverse and extensive set of baselines. We believe this comprehensive study strongly supports the practical significance of our work.
>
> >Architectural details underexplained: Some implementation choices (e.g., score function choices, transition structure) would benefit from more ablation or motivation.
>
> We appreciate the reviewer’s comment and would like to clarify these implementation choices. For the score network, we follow the architecture used in DDOM [1], with a few modifications: we employ a simple MLP with a hidden size of 1024 and use 4 layers with Swish activation, compared to DDOM’s original 2-layer ReLU setup. Regarding the transition structure, it is determined by the specific choice of the probabilistic bridge. In our case, we adopt the Brownian bridge as a practical instantiation of our general framework. The corresponding transition formulation, training procedure, and hyperparameter settings are provided in Appendix B.2.
>
> [1] Krishnamoorthy, Siddarth, Satvik Mehul Mashkaria, and Aditya Grover. "Diffusion models for black-box optimization." In International Conference on Machine Learning, pp. 17842-17857. PMLR, 2023.
>
> >How robust is ROOT to poor offline data coverage? Have you tested performance under varying dataset support quality?
>
> We evaluated ROOT’s efficacy for poor offline data coverage in the Few-Shot Experimental Design setting introduced by ExPT [2], where only 1% of the offline data is labeled and the remaining 99% is unlabeled. As shown in Table 4, ROOT performs effectively under this limited supervision scenario and even outperforms ExPT.
> Additionally, we conducted further experiments to assess ROOT’s robustness under varying dataset support quality. Specifically, we trained ROOT and other baselines using only the $p$% poorest-performing designs from the offline dataset. The results, shown in the table below, demonstrate that ROOT continues to outperform the baselines, even under these more challenging conditions. We however note that data-scarce setting is not the main focus of our work.
>
> | TFBind8 ($p$%) | ROOT           | GA             | COMs           | REINFORCE      | LTR            |
> |-|-|-|-|-|-|
> | 50          | **0.964 ± 0.015** | 0.580 ± 0.199  | 0.935 ± 0.052  | 0.915 ± 0.039  | 0.959 ± 0.022  |
> | 20          | **0.946 ± 0.045** | 0.480 ± 0.218  | 0.872 ± 0.085  | 0.917 ± 0.040  | 0.927 ± 0.033  |
> | 10          | **0.915 ± 0.019** | 0.559 ± 0.170  | 0.771 ± 0.128  | 0.913 ± 0.038  | 0.909 ± 0.034  |
>
> | Ant ($p$%) | ROOT           | GA             | COMs           | REINFORCE      | LTR            |
> |-|-|-|-|-|-|
> | 50      | **0.909 ± 0.012** | 0.394 ± 0.023  | 0.898 ± 0.035  | 0.317 ± 0.016  | 0.909 ± 0.042  |
> | 20      | **0.930 ± 0.023** | 0.663 ± 0.065  | 0.880 ± 0.027  | 0.261 ± 0.052  | 0.871 ± 0.059  |
> | 10      | **0.861 ± 0.051** | 0.619 ± 0.120  | 0.845 ± 0.041  | 0.281 ± 0.034  | 0.813 ± 0.026  |
>
> [2] Tung Nguyen, Sudhanshu Agrawal, and Aditya Grover. Expt: Synthetic pretraining for few-shot experimental design. Advances in Neural Information Processing Systems, 36:45856–45869, 2023.
>
>
>
> >Can you clarify the computational cost of ROOT compared to other diffusion-based energy-based approaches?
>
> We would like to analyze the detailed computational cost along with the running time of ROOT compared to another diffusion-based method, DEMO [3].
>
> Computational cost of ROOT is as follows:
> - For each training epoch: Fitting $n_f$ Gaussian processes requires $O(n_f n^3)$ where $n$ is the offline data size. Performing M gradient steps (querying the GP’s mean function) to generate $n_p$ synthetic points from each of $n_f$ functions requires $O(n_f M n_p n)$. Training the BBDM model requires T diffusion steps per sample, where each step involves a forward and backward pass through the score network $\epsilon_\theta$, giving a total cost of $O(T (f_\theta + b_\theta) n_f n_p)$, where $f_\theta, b_\theta$ denotes the cost of forward and backward, respectively.
> - For the whole training process, the overall cost is $O(E(n_f n^3 + n_f M n_p n + T (f_\theta + b_\theta) n_f n_p))$ where $E$ is the number of epochs.
> - For the inference process, the overall cost is $O(Q D f_\theta)$ where $Q$ and $D$ are the number of selected candidates and denoising steps.
>
> Therefore, the total computational cost of ROOT is:
> $O(E(n_f n^3 + n_f M n_p n + T (f_\theta + b_\theta) n_f n_p)) + O(Q D f_\theta)$
>
> Furthermore, we note that the most complexity term in ROOT's computational cost, GP fitting, can be further mitigated by using sparse GP techniques [4], which scales linearly in the number of inputs.
>
> --
>
> Correspondingly, the computational cost of DEMO is as follows:
> - Fitting a surrogate model: $O(E_f t_f n)$ where $E_f$ is the number of surrogate training epochs, $t_f$ is the cost of forward and backpropagation on the surrogate model.
> - Creating pseudo candidates by gradient ascent on the trained surrogate model: $O(Q M t_f)$
> - Training of diffusion prior: $O(E_d T (f_\theta + b_\theta) n)$ where $E_d$ is the number of diffusion training epochs.
> - Design Editing Process: $O(Q D f_\theta)$
>
> The total computational cost of DEMO is:
> $O(E_f t_f n) + O(Q M t_f) + O(E_d T (f_\theta + b_\theta) n) + O(Q D f_\theta)$
>
> Under the same hyper-paramters, model size and method for training and inference, ROOT might have theoretically higher computational cost than DEMO. This is however not critical as ROOT still scales well in all tested data benchmark compared to other baselines in Table 10 in the Appendix. As noted above, we can further reduce the complexity cost of ROOT via using sparse Gaussian processes in future work.
>
> Furthermore, ROOT empirically demonstrates significantly faster runtime compared to DEMO, as reported in the table below. This is because DEMO requires training both the surrogate model and the diffusion prior on the offline data for $E_f = 200$ and $E_d = 200$ epochs, respectively, compared to a single training phase of $E = 100$ epochs on small set of synthetic data of ROOT. Additionally, the running time gap also comes from the difference in training and inference method. While ROOT employs a fast DDPM-based training and only $D = 200$ steps of efficient DDIM sampling [5], DEMO use a slower SDE-based training and $D = 400$ sampling steps using a complex second-order Heun solver.
>
> | Method | Ant   | Dkitty | TFBind8 | TFBind10 |
> |-|-|-|-|-|
> | DEMO         | 668s | 1489s  | 1024s   | 1696s    |
> | ROOT         | 298s  | 297s   | 407s    | 575s    |
>
> We will update the above detailed computational analysis to the appendix.
>
> [3] Ye Yuan, Youyuan Zhang, Can Chen, Haolun Wu, Melody Zixuan Li, Jianmo Li, James J. Clark, and Xue Liu. Design editing for offline model-based optimization. Transactions on Machine Learning Research, 2025
>
> [4] Quinonero-Candela, Joaquin, and Carl Edward Rasmussen. "A unifying view of sparse approximate Gaussian process regression." Journal of machine learning research 6, no. Dec (2005): 1939-1959.
>
> [5] Song, Jiaming, Chenlin Meng, and Stefano Ermon. "Denoising diffusion implicit models." arXiv preprint arXiv:2010.02502 (2020).
>
> We hope that our responses have satisfactorily addressed your remaining concerns. If you have any further questions or points that require additional clarification, please feel free to let us know. We would be happy to continue the discussion and provide more details.

---

> > ### Comment · Reviewer_fMmR · 2025-08-04
> > **Reply to the rebuttal**
> >
> > Thank you for your detailed rebuttal and for taking the time to address my concerns, as well as for running additional experiments. I appreciate the clarifications and your effort to strengthen the submission. Therefore, I raised my score to 5 to further support the paper's acceptance. Best wishes to the authors.

---

> > > ### Author Response · Authors · 2025-08-05
> > > **Thank you for increasing rating!**
> > >
> > > Dear Reviewer fMmR,
> > >
> > > Thank you for reviewing our work and thoughtfully considering our rebuttal. We truly appreciate that our responses could address your concerns.
> > >
> > > We sincerely appreciate your recognition of our contribution and your decision to raise the score in support of acceptance.
> > >
> > > Best regards,
> > >
> > > The Authors

---

### Official Review · Reviewer_hCxA · 2025-07-03

**Clarity:** 3
**Significance:** 3
**Originality:** 2
**Rating:** 5
**Confidence:** 4

**Summary:**

This paper focuses on offline black-box optimization, which aims to maximize an objective function solely given an offline dataset. Previous works built mapping either from $x$ to $y$ or from $y$ to $x$, while this paper rethinks the optimization as from $x_{low}$ to $x_{high}$ (distributional translation as mentioned in the paper), which can be modeled by a probabilistic bridge model (initiated as a diffusion model). It utilizes Gaussian processes to derive synthetic data for diffusion training, then obtain the final solutions given solutions for the offline dataset after the model is trained.

**Questions:**

- Why do you conduct gradient ascent / descent on the mean functions of GP to obtain the synthetic data? As GP not only models the value but also quantifies the uncertainty. If the offline dataset also has limited coverage, only focusing on mean functions (without integrating the uncertainty of GP) could also lead to unfavorable data. Thus, I suggest comparing constructing synthetic data on UCB or LCB acquisition functions.
- In Introduction and Section 5, you mention that previous methods remain constrained by limited data coverage. Why does your method address this challenge? More discussions are needed.
- How does your proposed method perform under different data scarcities (not only the few-shot settings in Table 4)?

**Ethical Concerns:**

["NO or VERY MINOR ethics concerns only"]

**Final Justification:**

The proposed method is technically solid and intuitive. Other reviewers' concerns on the lack of theory is a common issue of this community. The thorough ablation studies examines the effectiveness of ROOT. Overall, I would recommend accepting this paper, which contributes a lot to offline optimization community.

**Limitations:**

See Weaknesses and Questions above. I am willing to increase my score once my concerns are addressed.

**Quality:**

3

**Strengths And Weaknesses:**

## Strengths

- The usage of diffusion models in offline optimization is novel.
- The modeling of mapping between $x$s **with different qualities** is intuitive.
- The experimental evaluations are comprehensive and clearly demonstrate the effectiveness of this paper.
- The implementation code is clear.

## Weaknesses

- Based on my comprehension, constructing a mapping between different $x$s is also been employed by recent works [1-2] (although they do not project from $x_{low}$ to $x_{high}$). Relevant discussion is needed.

## References

[1] From Function to Distribution Modeling: A PAC-Generative Approach to Offline Optimization. arXiv 2024.

[2] Design Editing for Offline Model-based Optimization. TMLR 2025.

---

> ### Author Rebuttal · Authors · 2025-07-31
>
> We would like to thank the reviewer for the detailed feedback and address the reviewer's questions as follows:
>
> >Based on my comprehension, constructing a mapping between different $x$s is also been employed by recent works [1-2] (although they do not project from $x_{low}$ to $x_{high}$). Relevant discussion is needed.
>
> We appreciate the reviewer’s suggestion and will include the discussion below of these relevant works in the related work section of the final version.
>
> Recent diffusion-based methods such as [1] and [2] also explore mappings between different input designs, albeit with different goals and mechanisms. DEMO [2] trains a diffusion model to approximate the distribution of offline data, which is then used to edit perturbations from high-value surrogate designs to safer alternatives. In contrast, ROOT introduces a probabilistic bridge framework that explicitly maps low-value inputs to high-value regimes. ROOT also significantly outperforms DEMO across multiple benchmarks, as reported in Table 1.
>
> The proposed method in [1] alternatively constructs a utility-weighted distribution by learning a data-dependent weight function, then trains a DDPM to simulate this distribution from an isotropic Gaussian, enabling direct sampling of high-value designs with a PAC guarantee that provably guards against poor average-case performance. In contrast, our low-to-high approach empirically simulates transitions from low- to high-value distributions through synthetic data generation. This strategy targets the best achievable average-case performance by progressively biasing generation toward high-value regions. Interestingly, the PAC-guided reweighting strategy from [1] could be integrated into our framework to enhance reliability, offering a principled mechanism to regularize the sampling trajectory and mitigate potential failure cases, especially in future problem domains with significantly higher complexity than the tested benchmarks. Although this integration is beyond the scope of the current work, we view it as a promising direction for future research.
>
> We sincerely thank the reviewer for introducing this valuable contribution to our attention. We will cite [1] in our revision.
>
> >Why do you conduct gradient ascent / descent on the mean functions of GP to obtain the synthetic data? As GP not only models the value but also quantifies the uncertainty. If the offline dataset also has limited coverage, only focusing on mean functions (without integrating the uncertainty of GP) could also lead to unfavorable data. Thus, I suggest comparing constructing synthetic data on UCB or LCB
>
> We appreciate the reviewer’s insightful comment. It is indeed true that relying solely on the mean function of a single GP trained on limited data may result in suboptimal synthetic data. However, our approach addresses this limitation by employing a large ensemble of GPs with diverse kernel hyperparameters, rather than a single model. This ensemble is constructed by uniformly sampling hyperparameters from a fixed range, which implicitly captures a broad spectrum of uncertainty within the offline dataset. This strategy is also motivated by ExPT (for few-shot offline optimization), which demonstrated strong empirical performance.
>
> We also agree that incorporating explicit uncertainty via acquisition functions like UCB or LCB is a promising alternative, especially in balancing exploration and exploitation. To evaluate this, we conducted a small-scale experiment replacing the GP’s mean function with UCB and LCB scores to generate synthetic data. The results, shown in the table below, indicate that while UCB and LCB perform reasonably well, they do not outperform our original approach based on the GP’s mean function. We believe this is because uncertainty has already been implicitly captured through a different mechanism, by sampling multiple mean functions from a population of posterior GPs trained on the same offline dataset using different kernel configurations.
>
> | Method | Ant           | TFBind8       |
> |--------|---------------|---------------|
> | UCB    | 0.964 ± 0.017 | 0.966 ± 0.017 |
> | LCB    | 0.949 ± 0.008 | 0.961 ± 0.025 |
> | ROOT   | **0.965 ± 0.014** | **0.986 ± 0.007** |
>
>
> >In Introduction and Section 5, you mention that previous methods remain constrained by limited data coverage. Why does your method address this challenge? More discussions are needed.
>
> In previous methods, performances are often limited by the quantity and quality of offline data, particularly when such datasets are sampled predominantly from low-value regions of the design space. Approaches such as forward modeling, inverse modeling, and learning search policies rely solely on the available offline data, which constrains their performance. In contrast, our approach addresses this challenge by generating synthetic data through Gaussian processes, effectively expanding the dataset as needed by producing additional samples (from similar functions). This aligns with the ultimate goal of offline optimization, which is to identify improved designs in regions surrounding the offline data where most functions consistent with the available evidence are also likely to agree. Empirical results demonstrate that ROOT successfully leverages this expanded synthetic data, achieving state-of-the-art performance across various benchmarks. We will add this clarification to the key discussion at lines 42-57 in the introduction.
>
> >How does your proposed method perform under different data scarcities (not only the few-shot settings in Table 4)?
>
> We conducted additional experiments to evaluate the performance of our method under varying levels of data scarcity beyond the few-shot setting in Table 4. Specifically, we simulated limited data coverage by using only the $p$% poorest-performing designs from the offline dataset. As shown in the table below, ROOT consistently outperforms other baselines across different levels of data scarcity, demonstrating its robustness even in more challenging, low-coverage scenarios. We note that data-scarcity setting is however not our focus in this work.
>
> | TFBind8 ($p$%) | ROOT           | GA             | COMs           | REINFORCE      | LTR            |
> |-|-|-|-|-|-|
> | 50 %         | **0.964 ± 0.015** | 0.580 ± 0.199  | 0.935 ± 0.052  | 0.915 ± 0.039  | 0.959 ± 0.022  |
> | 20 %         | **0.946 ± 0.045** | 0.480 ± 0.218  | 0.872 ± 0.085  | 0.917 ± 0.040  | 0.927 ± 0.033  |
> | 10 %        | **0.915 ± 0.019** | 0.559 ± 0.170  | 0.771 ± 0.128  | 0.913 ± 0.038  | 0.909 ± 0.034  |
>
> | Ant ($p$%) | ROOT           | GA             | COMs           | REINFORCE      | LTR            |
> |-|-|-|-|-|-|
> | 50 %     | **0.909 ± 0.012** | 0.394 ± 0.023  | 0.898 ± 0.035  | 0.317 ± 0.016  | 0.909 ± 0.042  |
> | 20 %     | **0.930 ± 0.023** | 0.663 ± 0.065  | 0.880 ± 0.027  | 0.261 ± 0.052  | 0.871 ± 0.059  |
> | 10 %     | **0.861 ± 0.051** | 0.619 ± 0.120  | 0.845 ± 0.041  | 0.281 ± 0.034  | 0.813 ± 0.026  |
>
> We hope our responses have fully addressed your remaining concerns, and we would sincerely appreciate your consideration in raising your score if no further issues persist. If you have any further questions or points that require additional clarification, please feel free to let us know. We would be happy to continue the discussion and provide any additional details.

---

> > ### Comment · Reviewer_hCxA · 2025-08-02
> >
> > Thanks for your detailed responses! Most of my concerns are addressed. I specifically appreciate the authors' discussion on ROOT's ability to address the data scarcity challenge, where synthetic data might unlock the potential of optimization ability on few-shot dataset. From my perspective, it is quite essential for the community, as synthetic data has been proven useful in other scenarios, e.g., tabular machine learning and large language model. Although other reviewers raise concerns on the lack of theory, this paper is technically solid with lots of ablation studies. Thus, I decide to raise my score to 5.
> >
> > I additionally have a minor concern of this community. In offline optimization, ones rely on Design-Bench to evaluate the methods, where the offline dataset only covers the poorest region of the collected full dataset. However, in many real-world scenarios, the offline dataset would not cover the poorest region. Thus, a more solid evaluation for this community is necessary.

---

> > > ### Author Response · Authors · 2025-08-02
> > > **Thank you for the increased rating!**
> > >
> > > Dear Reviewer hCxA,
> > >
> > > We are very glad that the rebuttal has addressed your concern.
> > >
> > > Thank you for recognizing our contribution!
> > >
> > > Best regards,
> > >
> > > Authors

---

### Official Review · Reviewer_zEZb · 2025-07-05

**Clarity:** 3
**Significance:** 3
**Originality:** 3
**Rating:** 5
**Confidence:** 3

**Summary:**

The paper proposes ROOT, an offline black‑box optimization framework that recasts the task as a distributional translation problem. A probabilistic bridge, inspired by diffusion but conditioned on both source and target points, is trained on synthetic low‑ and high‑value design pairs generated from an ensemble of Gaussian‑process posterior means fitted to the offline data. Once trained, the bridge is run backward from the best offline samples to produce improved candidates. Experiments on four Design‑Bench tasks and three RNA inverse‑folding tasks report new state‑of‑the‑art results and extensive ablations support each design choice.

**Questions:**

Please clarify the computational cost of fitting eight hundred GPs versus a single large GP ensemble and how this scales beyond ten thousand offline points; discuss whether fewer but higher‑rank kernels could suffice.

Explain how ROOT would handle objectives with strong measurement noise, since synthetic labels come from deterministic GP means.

Provide results on at least one truly high‑dimensional continuous task (e.g., 1000‑dim design) to assess scalability.

Describe how sensitive performance is to the choice of Brownian‑bridge parameters and whether other kernel choices were tried.

A detailed comparison with very recent diffusion‑based baselines such as DiGBO or DM‑BO would strengthen the empirical claim.

**Ethical Concerns:**

["NO or VERY MINOR ethics concerns only"]

**Final Justification:**

Thank you for the thorough rebuttal. The new experiments on high-dimensional (Hopper) and noisy tasks addressed my primary concerns regarding the evaluation. The clarifications on computational cost were also helpful. After reviewing the strong responses provided to all reviewers and seeing the consensus, it is clear the paper is strong. You have fully addressed my concerns, and I am happy to raise my score to 5.

**Limitations:**

The authors list data and compute limitations and note lack of theory, but they should also acknowledge that GP‑derived synthetic data may embed biases of the initial kernel, potentially restricting exploration.

**Quality:**

3

**Strengths And Weaknesses:**

Quality: Empirical work is thorough, with eight independent runs, multiple percentiles, and ablations on step size, GP ensemble size, and data budgets. However, the method is empirically driven; there is no theoretical guarantee that the learned bridge preserves function maxima, and some critical implementation details (e.g., GP kernel hyper‑parameter grids, training compute) are only in the appendix.

Clarity: The core idea is clearly motivated and the probabilistic bridge derivation is explicit, yet the paper is very long, and key intuition is buried in implementation, which may hinder reproducibility.

Significance: Achieving uniform gains across both continuous robot morphology and discrete sequence design benchmarks suggests practical impact, but the community still lacks evidence that gains transfer to harder high‑dimensional or noisy settings.

Originality: Framing offline optimization explicitly as distributional translation via a bridge that conditions on both endpoints is novel, though it builds heavily on recent diffusion‑based optimizers and GP‑assisted pretraining.

---

> ### Author Rebuttal · Authors · 2025-07-31
>
> We thank the reviewer for the positive rating and detailed feedback. We address the reviewer's questions as follows:
> > there is no theoretical guarantee that the learned bridge preserves function maxima
>
> We clarify that given sparse data, there exist infinitely many functions consistent with the observed samples, and their optima can differ substantially. As a result, recovering the true maximum is not the goal of offline optimization. Instead, the focus is on discovering improved designs in regions near the observed data, where most consistent functions are likely to agree.
>
> Our method is designed with this goal. By learning a probabilistic bridge from low- to high-value regions using simulated data from synthetic functions sampled around the oracle, we aim to discover inputs that likely yield better outcomes within this region that encode the oracle's plausible behaviors.
>
> We also agree that theoretical guarantees for the learned bridge consistently mapping to high-value regions would be valuable. However, this relates to broader open questions in generative modeling, where such guarantees are still largely lacking. Most offline optimization methods also lack guarantees for discovering improved designs. Deriving such results is therefore nontrivial and beyond the current scope, but we see it as a valuable direction for future work.
> > some critical implementation details (e.g., GP kernel hyper‑parameter grids, training compute) are only in the appendix.
>
> We will add the following details to the “Hyper-parameter Configuration” paragraph in Section 4.1:
>
> For each baseline, we adopt the optimized settings from the original papers. For GP kernel hyper-parameters in our data generation, we sample lengthscales $l_s$ and variances $\sigma_s^2$ uniformly from $[l_0 − \delta, l_0 + \delta]$ and $[\sigma_0^2 - \delta, \sigma_0^2 + \delta]$, with $l_0 = \sigma_0^2 = 1.0$ for continuous tasks and 6.25 for discrete tasks, and $\delta = 0.25$. We use $M=100$ gradient steps with step sizes 0.001 (continuous) and 0.05 (discrete). Additional data generation details are in App. B.1. For training the Probabilistic Bridge model, we use a Brownian Bridge diffusion process with the Adam optimizer over $E=100$ epochs and $n_g=800$ synthetic functions, running on a single NVIDIA A100-80GB GPU. More training details are provided in App. B.2.
> > yet the paper is very long, and key intuition is buried in implementation, which may hinder reproducibility..
>
> We would like to point out that the core intuition of our work is outlined in lines 51–59 (Section 1) and further elaborated at the beginning of Section 3. We will highlight these more explicitly in our revision.
>
> To summarize, we cast offline optimization as a translation task from a low-value source distribution to a high-value target distribution. To facilitate this, we introduce the probabilistic bridge concept, which identifies localized translation examples by conditioning on both source and target contexts (Section 3.1.1). These examples are then utilized to train a global, target-agnostic transformation flow that generalizes beyond the observed data (Section 3.1.2).
>
> We provide a concrete example in Section 3.2, which illustrates how a Brownian bridge can instantiate our framework, with implementation details in Appendix B.2. Its derivation is deferred to the appendix because it is procedural rather than insightful. We can instantiate other bridges using the same procedural derivation (via deriving the underpinning transition via conditional Gaussian rule on Gaussian processes) with different kernel choices. For example, we have presented another practical instantiation with the Ornstein-Uhlenbeck bridge (see Appendix E in the updated Appendix) in the Supplementary Material.
>
> > the community still lacks evidence that gains transfer to harder high‑dimensional or noisy settings
>
> We appreciate the reviewer’s broader observation that the community still lacks strong evidence of gains transferring to high-dimensional or noisy settings. We agree that this remains an important challenge of the offline optimization community. By connecting offline optimization with the broader generative modeling toolkit, we introduce a principled and data-driven bridge construction mechanism generalizing diffusion processes, which have demonstrated robustness and stability across diverse domains. This connection opens a promising pathway toward more scalable and resilient offline optimization, particularly in high-dimensional or noisy settings. To provide further insight, we have conducted additional experiments evaluating the efficacy of ROOT in both high-dimensional and noisy scenarios, as detailed in our responses below.
>
> > Please clarify the computational cost of fitting eight hundred GPs versus a single large GP ensemble and how this scales beyond ten thousand offline points; discuss whether fewer but higher‑rank kernels could suffice.
>
> In our setting, the 800 GPs are not trained via hyperparameter optimization but use instead kernel parameters sampled from a prior, enabling efficient posterior mean computation with minimal overhead. In contrast, learning a GP ensemble requires learning a posterior over kernel hyperparameters, often through marginal likelihood optimization, which can be significantly more expensive in both compute and memory. In particular, we have explored 2 ensemble-based approaches that underperformed compared to ROOT and incurred higher computational costs (Appendix C.6).
>
> Regarding the suggestion to use fewer but higher-rank kernels, the RBF kernel already corresponds to an infinite-dimensional feature space, so increasing its rank is not applicable. However, exploring alternative kernel families with richer inductive biases is an interesting direction for future work.
>
> To scale beyond ten thousand offline points, our framework supports direct use of sparse GP approximations, which scale linearly with data size. Additionally, individual GPs can be trained on data subsets to further improve scalability.
>
> Overall, while using a full GP ensemble with learned hyperparameters is feasible, it would significantly increase computational overhead. Since our approach already performs well with much lower cost, we prioritized scalability and efficiency in the current scope. Extending the method with more expressive GP ensembles remains a promising direction for future work.
>
> >Explain how ROOT would handle objectives with strong measurement noise, since synthetic labels come from deterministic GP means.
>
> Although ROOT uses synthetic labels from deterministic GP means, it remains robust to strong measurement noise. This is due to the use of a diverse GP ensemble with varied hyperparameters, which implicitly captures a wide range of uncertainties present in the offline dataset. Sampling from this ensemble produces synthetic data that reflects variability in the objective, helping the bridge model generalize more effectively. To validate this, we ran additional experiments on TFBind8 with label perturbed by Gaussian noise $N(0,\epsilon)$. As shown in the table below, ROOT maintains strong performance and outperforms other baselines across noise levels, demonstrating its resilience to noise.
>
> |$\epsilon$|0.01|0.1|0.2|
> |-|-|-|-|
> |GA|0.967±0.015|0.970±0.006|0.963±0.010|
> |COMs|0.956±0.024|0.925±0.034|0.952±0.013|
> |REINFORCE|0.947±0.030|0.933±0.025|0.930±0.042|
> |ROOT|**0.969±0.016**|**0.971±0.007**|**0.965±0.014**|
>
> >Provide results on at least one truly high‑dimensional continuous task (e.g., 1000‑dim design) to assess scalability.
>
> In the Design-Bench benchmark, the Hopper task represents a high-dimensional continuous task with 5,126 input dimensions. Although prior work has noted a highly noisy and inaccuracy oracle function for this task, we conducted a small experiment to evaluate ROOT’s performance on Hopper as a test of scalability. As shown in the table below, ROOT outperforms some representative baselines, demonstrating its ability to effectively handle high-dimensional design spaces.
>
> |Method|Hopper Controller|
> |-|-|
> |GA|-0.068±0.001|
> |MINs|0.267±0.350|
> |Reinforce|-0.009±0.067|
> |**ROOT**|**0.541±0.042**|
>
> >Describe how sensitive performance is to the choice of Brownian‑bridge parameters and whether other kernel choices were tried.
>
> While we primarily tune the hyperparameters of the synthetic data generation process as detailed in Appendix C.7, the Brownian bridge parameters we use are either inherited from the Brownian Bridge Diffusion Model (BBDM) or set to standard values (e.g., learning rate, batch size, dropout). To further assess the ROOT's sensitivity, we conducted an additional experiment on ANT with some BBDM hyperparameters. As shown in the tables below, ROOT's performance remains stable, indicating robustness to the choice of Brownian bridge parameters.
> |Dropout|0.05|0.1|0.15|0.2|
> |-|-|-|-|-|
> |ROOT|0.957±0.020|0.960±0.014|**0.965±0.014**|0.957±0.019|
>
> |Batch size|32|64|128|
> |-|-|-|-|
> |ROOT|0.956±0.014|**0.965±0.014**|0.955±0.008|
>
> Additionally, we experimented with the Matern kernel for the GP-based synthetic data generation. The results indicate that the commonly used RBF kernel consistently yields better performance, supporting our decision to use it as the default.
> |GP kernel|Ant|TFBind8|
> |-|-|-|
> |Matern|0.966±0.013|0.848±0.055|
> |RBF|0.965±0.014|0.986±0.007|
>
> >A detailed comparison with very recent diffusion‑based baselines such as DiGBO or DM‑BO would strengthen the empirical claim.
>
> We would greatly appreciate if the reviewer could provide us with paper titles for DiGBO and DM-BO because we cannot find these methods within the context of offline optimization. Our online searches did not return any matching results.
>
> We hope our responses have fully addressed your concerns. If you have any further questions or points that require additional clarification, please feel free to let us know. We would be happy to continue the discussion and provide any additional details.

---

> > ### Comment · Reviewer_zEZb · 2025-08-04
> >
> > Thank you for the thorough rebuttal. The new experiments on high-dimensional (Hopper) and noisy tasks addressed my primary concerns regarding the evaluation. The clarifications on computational cost were also helpful. After reviewing the strong responses provided to all reviewers and seeing the consensus, it is clear the paper is strong. You have fully addressed my concerns, and I am happy to raise my score to 5.

---

> > > ### Author Response · Authors · 2025-08-05
> > > **Thank you for increasing rating!**
> > >
> > > Dear Reviewer zEZb,
> > >
> > > Thank you for taking the time to review our work and for thoughtfully considering our rebuttal. We’re truly grateful that our responses helped address your concerns.
> > >
> > > We sincerely appreciate your recognition of our contribution and your increased score in favor of acceptance.
> > >
> > > Best regards,
> > >
> > > The Authors

---

### Note · Authors · 2025-08-16

Dear AC and Reviewers,

We sincerely thank you for the detailed feedback and the thorough review process.

In our rebuttal, we addressed the following key points:

- We provided further details on the computational cost of our proposed method and added empirical results demonstrating ROOT’s robustness under strong measurement noise and in extremely high-dimensional tasks.

- We supplied additional implementation details together with new results showing ROOT’s robustness to the choice of Brownian bridge parameters.
- We presented new results demonstrating ROOT’s efficiency under varying levels of data scarcity.
- We addressed specific questions regarding strategies for limited-data challenges and performance on real-world tasks, such as biological sequence design.

Overall, we are happy that our rebuttal has sufficiently addressed the reviewers’ questions and contributed to an overall increase in ratings. In particular, we are grateful to reviewers zEZb, hCxA, and fMmR for raising their ratings to acceptance (5), and to reviewer tDV4 for maintaining the acceptance rating (5).

---

### Decision · Program_Chairs · 2025-09-17

**Decision:**

Accept (spotlight)

**Comment:**

This paper addresses black-box optimization with limited offline data and proposes a novel approach that formulates the problem as a distributional translation task. The method combines surrogate optimization with inverse model estimation. Specifically, the authors train a probabilistic bridge—conceptually inspired by diffusion but conditioned on both source and target points—using synthetic low- and high-value design pairs generated from an ensemble of Gaussian-process posterior means fitted to the offline data. Once trained, this bridge is run backward from the best offline samples to generate improved candidates.

Most reviewers agreed that the paper merits acceptance. After carefully reviewing the rebuttal and subsequent discussion, I concur. Nevertheless, I encourage the authors to address the reviewers’ feedback in the final version of the paper, with particular emphasis on:

Including experiments that consider high-dimensional and noisy data.

Citing the missing related works and providing empirical comparisons with some of them, as suggested by the reviewers.